# SYLLABLELM: LEARNING COARSE SEMANTIC UNITS FOR SPEECH LANGUAGE MODELS

**Alan Baade, Puyuan Peng, David Harwath**
Department of Computer Science
The University of Texas at Austin
`abaade@utexas.edu`

## ABSTRACT

Language models require tokenized inputs. However, tokenization strategies for continuous data like audio and vision are often based on simple heuristics such as fixed sized convolutions or discrete clustering, which do not necessarily align with the semantic structure of the data. For speech in particular, the high resolution of waveforms (16,000 samples/second or more) presents a significant challenge as speech-based language models have had to use several times more tokens per word than text-based language models. In this work, we introduce a controllable self-supervised technique to merge speech representations into coarser syllable-like units while still preserving semantic information. We do this by 1) extracting noisy boundaries through analyzing correlations in pretrained encoder losses and 2) iteratively improving model representations with a novel distillation technique. Our method produces controllable-rate semantic units at as low as 5Hz and 60bps and achieves SotA in syllabic segmentation and clustering. Using these coarse tokens, we successfully train SyllableLM, a Speech Language Model (SpeechLM) that matches or outperforms current SotA SpeechLMs on a range of spoken language modeling tasks. SyllableLM also achieves significant improvements in efficiency with a 30x reduction in training compute and a 4x wall-clock inference speedup. Our code and checkpoints are available at `https://www.github.com/alanbaade/SyllableLM`.

## 1 INTRODUCTION

Learning to generate speech solely from listening to spoken language is a fundamental task in speech processing. It requires abstracting beyond the underlying acoustics of speech into phones, syllables, words, and sentences to process correlations across long ranges of time. But while current textual language models (Touvron et al., 2023; Zhang et al., 2022; Brown et al., 2020) can compose highly realistic text, language models on spoken language still struggle to output semantically meaningful speech. An increasing focus has coalesced around Generative Spoken Language Modeling (GSLM) (Lakhotia et al., 2021), which sets out to achieve this goal.

The most successful approaches to GSLM are Transformer Decoder Language Models (Vaswani et al., 2017) such as AudioLM (Borsos et al., 2023) and TWIST (Hassid et al., 2023). These Speech Language Models (SpeechLMs) operate on discrete tokens output by self-supervised learning (SSL) based encoder models (Hsu et al., 2021; Chung et al., 2021). However, existing SSL tokenizations predominantly capture phonetic information (Choi et al., 2024) at a high temporal resolution of 25-50 tokens per second, much greater than the typical human speaking rate of 2-5 words per second. This large number of tokens substantially impairs both training and inference speed (Hassid et al., 2023), and it is unclear whether modeling speech with such a high granularity harms semantic understanding.

Very recently, there has been significant progress in extracting coarser speech unit representations from raw audio. In particular, SD-HuBERT (Cho et al., 2024) finetunes HuBERT (Hsu et al., 2021) with a DINO-like distillation objective, and VG-HuBERT (Peng & Harwath, 2022; Peng et al., 2023) uses a contrastive loss against cross-modal visual inputs. We continue and significantly improve upon this line of research, resulting in high quality speech tokens (units) suitable for SpeechLMs that exhibit a temporal resolution roughly corresponding to syllables. Specifically, we demonstrate

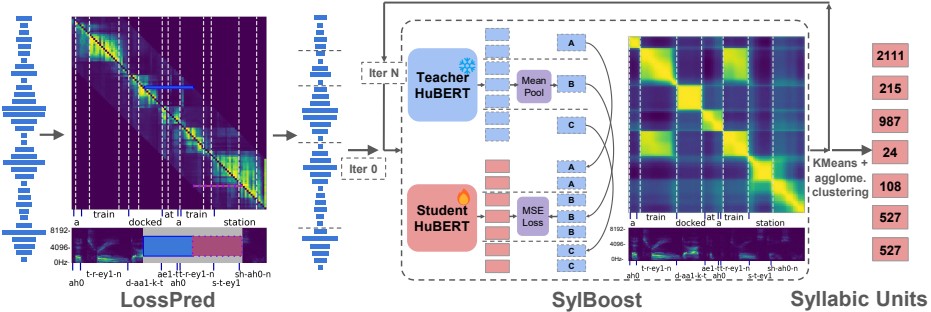

Figure 1: Left-Top: The loss prediction matrix $C$, where brighter is higher likelihood placed on the teacher label. A time-aligned transcript is on the bottom, and predicted cluster unit boundaries span vertically as dashed-lines. Left-Bottom: A Mel-Spectrogram of the input waveform with an example masked timespan in gray. The losses on tokens at timesteps covered by the solid blue and dotted red spans are mapped to their corresponding rows and columns in $C$ as described in Section 3.1. Right: Visual of SylBoost. We train a student to match intermediate teacher features pooled over regions generated by pseudo-syllable-boundaries. We use a min-cut algorithm on the feature self-similarity matrix to extract boundaries, and then apply K-Means and Agglomerative clustering to obtain discrete units.

significant improvements in textual reconstruction from low-bitrate units of SSL models, reducing the word-error-rate (WER) from 37% using SD-HuBERT units to 7%, and more than halving realized bitrate of previous SpeechLM units from 175bps to as low as 60bps. We additionally find that our units correlate strongly with syllables both in boundary detection and in cluster quality.

Furthermore, we evaluate the effects of training SpeechLMs on these new units and obtain state-of-the-art results across a wide-variety of metrics, competitive with or outperforming AudioLM (350M parameters), all TWIST model sizes (125M-13B parameters), and the newly released Moshi-7B (Défossez et al., 2024) with significantly fewer parameters, lower training time, and faster inference speed. Our contributions are as follows:

1. We propose an unsupervised algorithm named LossPred that reveals noisy syllable-like segmentation of unannotated speech signals by analyzing the loss of a pretrained self-supervised model (e.g. HuBERT) across different masking spans.
2. We propose a method named SylBoost that sharpens the feature space of self-supervised speech models by bootstrapping from LossPred syllable-like segmentations. SylBoost achieves SotA unsupervised syllabic segmentation, categorization, and low-bitrate unit-to-audio resynthesis.
3. Using quantized SylBoost units as a basis for tokenization, we train SyllableLM, a SpeechLM that outperforms or matches prior approaches on a range of tasks while being 30x faster to train, 4x faster for inference, and having a 2.5x reduction in unit bitrate.

## 2 RELATED WORK

**Self-Supervised Encoder Models**   There has been a great amount of work in learning high-level representations from data by reconstructing corrupted inputs across speech (Baevski et al., 2020; Hsu et al., 2021; Baevski et al., 2023), audio (Gong et al., 2021), text (Devlin et al., 2019; Clark et al., 2020), and vision (Caron et al., 2021; He et al., 2021). To navigate the lack of simple discrete targets in speech, much work has been placed in finding high-quality targets, such as iterative clustering (Hsu et al., 2021) and by predicting the representations of a teacher network based on a running average of student model weights (Baevski et al., 2022; 2023). An alternate but similar line of work has been placed into learning low-bitrate units for the task of resynthesis (Défossez et al., 2023; Zeghidour et al., 2021; Yang et al., 2023a; Kumar et al., 2023; Zhang et al., 2023; Du et al., 2023), which include losses focused on reconstruction and use an information bottleneck to enforce compression.

**Applications of Neural Codecs**    The discrete units generated by these self-supervised encoders are versatile and fundamental to much of the recent progress in speech research such as Text-To-Speech (Wang et al., 2023; Ju et al., 2024; Song et al., 2024; Peng et al., 2024), joint audio-text foundation models (Yang et al., 2023b; Chou et al., 2023; Nguyen et al., 2024), unsupervised speech recognition (Baevski et al., 2021), discrete unit resynthesis (Polyak et al., 2021; Défossez et al., 2023; Zeghidour et al., 2021), text-to-audio (Kreuk et al., 2022; Agostinelli et al., 2023; Copet et al., 2023), and generative spoken language modeling (Borsos et al., 2023; Hassid et al., 2023; Lakhotia et al., 2021). Each of these methods operates on audio units exclusively greater than or equal to 25Hz, which has been a frequently cited area for future work to improve on (Hassid et al., 2023). Recent work (Elkahky et al., 2023) has also explored training speech encoder models with coarser units as targets.

**Extracting Semantic Units from Raw Data**    Additionally relevant to our work are several approaches, particularly in vision and audio, that generate emergent semantic clusterings from self-supervised transformer (Vaswani et al., 2017) models. In particular, the DINO approach in Caron et al. (2021) observes object representations in attention maps through student-teacher distillation. Similar techniques have been also applied to audio to discover emergent syllable boundaries (Cho et al., 2024; Peng et al., 2023). These behaviors can vary heavily with small changes in pretraining strategy as explored in Darcet et al. (2024). Merging similar features has also been shown to produce significant vision model speedups such as in Bolya et al. (2023). Most similar to our work, Algayres et al. (2023) extracted coarse continuous representations for GSLM, however these results trail behind language modeling approaches.

## 3    LEARNING SELF-SUPERVISED, SYLLABLE-LIKE REPRESENTATIONS FROM RAW SPEECH

In this section, we describe the bootstrapping process by which we extract low-bitrate speech units. We first describe LossPred, our algorithm to analyze outputs of self-supervised speech model loss functions to generate initial syllable-like unit boundaries. Following this, we define SylBoost, a procedure to iteratively refine these boundaries with student-teacher distillation. We also propose a new algorithm for the efficient extraction of boundaries from feature self-similarity matrices to fix the bottleneck slowing down VG-HuBERT and SD-HuBERT extraction.

### 3.1    LOSSPRED: EXTRACTING SYLLABLE-LIKE SEGMENTATION FROM RELATIONS IN HUBERT'S LOSS

HuBERT has previously been shown to learn phone-like units with its K-Means clusterings (Hsu et al., 2021) which have formed the basis of subsequent works on GSLM and unsupervised ASR (Baevski et al., 2021; Lakhotia et al., 2021; Hassid et al., 2023). However, other work (Pasad et al., 2023; 2024) has shown that the representations learned by these models also correlate with higher level structure such as words, despite these structures not immediately appearing during clustering. Our goal in this section is to propose a method that can be applied to a pretrained HuBERT model in order to automatically extract unit boundaries at the level of syllables or words rather than phones. Although we apply our method to HuBERT, we expect that it could also be applied to other SSL speech models that utilize a similar loss function such as WavLM (Chen et al., 2021) or wav2vec2.0 (Baevski et al., 2020). The crucial commonality between these models is that they all utilize a masked language modeling (MLM) training objective, whereby input speech tokens are randomly masked and the model is trained to predict the masked inputs conditioned on the unmasked inputs.

We ground our intuition for LossPred with the following thought experiment: if the input tokens corresponding to an entire word were replaced with mask tokens, we would expect the HuBERT model loss at these timesteps to be relatively high, as HuBERT would have to jointly predict word identity and the underlying acoustics to predict the missing span. On the other hand, if only the latter portion of a word were masked out, infilling this masked region given the word prefix may be easier by comparison. With this, if we iteratively shift a contiguous mask over a span of tokens and look at the loss, we would suspect to see a strong decrease in the loss throughout the timesteps corresponding to a masked semantic unit (word, syllable, or otherwise) as the beginning or end of the unit was partially revealed to the model.

LossPred operates using two frozen off-the-shelf HuBERT models, a student model and a teacher model, where the student model has been optimized to predict the quantized representations of the teacher model at layer $L$, and the teacher model comes from the prior iteration of pretraining as described in Hsu et al. (2021). Formally, given an input waveform $W$, we extract the teacher labels by passing $W$ into the teacher model and then quantizing the representations at layer $L$ with K-Means. We denote these teacher labels as $Y_{\{1...T\}}$, where $T$ is the number of timesteps. As done during HuBERT pretraining, we give the student model a corrupted version of $W$ where outputs from student CNN feature extraction at select timesteps are replaced with the HuBERT 'mask' embedding. We denote these student CNN features as $X^M_{\{1...T\}}$ where $M = \{t_1, \ldots t_m\}$ is a set of masked out timesteps. Following the pretext task, the student then predicts the teacher labels at masked timesteps $Y_t, t \in M$ conditioned on $X^M$ and is evaluated using a cross-entropy loss:

$$\mathcal{L}_{\text{HuBERT}} := -\log p(Y_t \mid X^M) \tag{1}$$

We define a loss prediction matrix $C$, shown in Figure 1, which looks at the losses of the student model as we move a contiguous sliding window of $s$ timesteps to mask. Specifically, $C$ models the raw probabilities of the losses where the column is the timestep being predicted and the row represents the closest unmasked token.

$$C_{r,c} \in \mathbb{R}^{T \times T}_+ = \begin{cases} p(Y_c \mid X^M) \mid M = \{r+1, r+2, \ldots r+s\} \cap [1, T] & \text{if } r < c, |r-c| \le \lfloor \frac{s}{2} \rfloor, \\ p(Y_c \mid X^M) \mid M = \{r-1, r-2, \ldots r-s\} \cap [1, T] & \text{if } r > c, |r-c| \le \lfloor \frac{s}{2} \rfloor, \\ 0 & \text{otherwise.} \end{cases} \tag{2}$$

We separately calculate the upper and lower triangles of $C$, relating to the closest observed waveform being before the mask and after the mask respectively. For simplicity, we assume here that the losses in the first half of the mask span are uncorrelated with information after the mask and vice versa. In the upper triangle, each entry $C_{r,c}$ at row $r$ column $c$ is equal to $p(Y_c \mid X^M)$ given that the mask span in $X^M$ starts just after time $r$. Inversely, in the lower triangle, $C_{r,c}$ is equal to $p(Y_c \mid X^M)$ given that the mask span ends just before time $r$. We use a span size $s = 50$ corresponding to 1 second as this duration is long enough to mask the majority of spoken words. We calculate the upper triangle using the first 25 timesteps of the mask span, and the lower triangle using the last 25.

To extract $k$ semantic regions with boundaries $B = \{b_1 = 1 < b_2 < \ldots < b_{k+1} = T+1\}$ from $C$, we use the min-cut algorithm from Shi & Malik (2000) discussed further in Peng et al. (2023), treating $C$ as the input feature-similarity matrix:

$$B := \operatorname*{arg\,max}_{\{b_1=1<b_2\ldots<b_{k+1}=T+1\}} \sum_{t=1}^{k} \frac{\displaystyle\sum_{i,j=b_t}^{b_{t+1}-1} C_{i,j}}{\displaystyle\sum_{i=b_t}^{b_{t+1}-1} \sum_{j=1}^{T} (C_{i,j} + C_{j,i}) - \sum_{i,j=b_t}^{b_{t+1}-1} C_{i,j}} \tag{3}$$

By choosing $k$ to be proportional to the length of the utterance, we can control the sample rate of our boundaries. We explore modifying this parameter in-depth throughout our experiments. We find that semantic units extracted by LossPred tend to be syllable-like (both via inspection, and also confirmed experimentally in our segmentation and clustering experiments) and so we focus on syllable-resolution units for the rest of the paper.

LossPred is expensive to run due to using one forward pass for each sliding window location. To make this efficient, we extract multiple masked spans simultaneously with a gap between spans of three seconds. This results in roughly 200 forward passes of the student model to calculate $C$ on an arbitrarily-sized audio. Fortunately, we only ever need to run LossPred once to initialize SylBoost training (described in 3.2), so this does not affect later semantic unit extraction speed.

### 3.2 SYLBOOST: BOOTSTRAPPING PESUDO-SYLLABIC UNITS WITH ITERATIVE DISTILLATION

Given the initial boundaries predicted by LossPred, we follow the paradigm of noisy-student-teacher learning (Xie et al., 2020) to extract semantic representations. Our goal is to "sharpen" the syllabic organization in the feature space of an input student model, as seen in Figure 2. We choose a

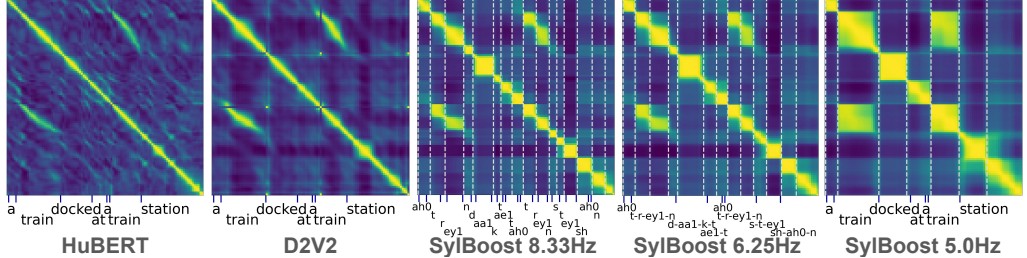

Figure 2: Qualitative results on SylBoost controllability for boundary detection. We plot the feature similarity matrix $A$, described in 3.2 for HuBERT, Data2Vec2, and SylBoost on Data2Vec2 when trained at different unit rates. The number of cuts $k$ is selected dynamically as described in 5.2.

pretrained HuBERT (Hsu et al., 2021) or Data2Vec2 (Baevski et al., 2023) to initialize our SylBoost student and teacher models, with the teacher model parameters held constant.

The SylBoost procedure is depicted in Figure 1. For a set of hypothesized speech segment boundaries $B = \{b_1 = 1 < b_2 < \ldots < b_{k+1} = T + 1\}$, we group together all temporal tokens between two boundaries into disjoint groups. $G_i = \{t \mid b_i \le t < b_{i+1}\}$. For notation, we let $H_t$ be the index of the group that contains $t$: $t \in G_{H_t}$. We apply the SylBoost loss (Equation 4) to model features at layer $L$, which we select based on syllabic correlation as explored in detail in Peng et al. (2023); Pasad et al. (2023; 2024). This results in student features $X^{(L)}_{\{1\ldots T\}}$ and teacher features $Y^{(L)}_{\{1\ldots T\}}$. The loss is then the mean squared error between the student features $X_t^{(L)}$ at each timestep and the mean of the teacher features in the timestep's corresponding group:

$$\mathcal{L}_{\text{SylBoost}} \coloneqq \frac{1}{T} \sum_{t=1}^{T} \left( X_t^{(L)} - \frac{1}{|G_{H_i}|} \sum_{s \in G_{H_i}} Y_s^{(L)} \right)^2 \tag{4}$$

After training, SylBoost results in a student model with sharpened latent features $X_t^{(L)}$ from which we calculate a matrix $A$ of pairwise frame distances where element $A_{i,j}$ equals the $L_2$ distance between the features $X_i^{(L)}$ and $X_j^{(L)}$ as depicted in Figure 2. We then extract boundaries from $A$ using a cut algorithm described in the next section (Sec. 3.3). With this, we can generate new pseudolabel boundaries and run SylBoost again to extract even better boundaries, which we perform twice. Finally, we extract discrete clusters by mean pooling the features at layer $L$ over each SylBoost boundary group, applying K-Means to produce a large number of clusters, and then using Agglomerative Clustering to merge similar clusters into a final desired number of units.

## 3.3 Efficient Extraction of Unit Boundaries for SylBoost

To extract boundary indices from learned feature representations Peng et al. (2023) proposed adapting the mincut approach in Malioutov & Barzilay (2006). However, for speech this approach is slow in practice and difficult to parallelize, bottlenecking our ability to extract boundaries in bulk across the large corpora necessary for downstream language modeling. Inspired by the SylBoost objective, we propose a more efficient approach for extraction: given $k + 1$ potential boundaries, we seek to choose groups that minimize the sum of the distances from each unit to the mean of its assigned group:

$$B \coloneqq \operatorname*{arg\,min}_{\{b_1=1<b_2\ldots<b_{k+1}=T+1\}} \sum_{i=1}^{k} \sum_{j=b_i}^{b_{i+1}-1} \left( X_j^{(L)} - \frac{1}{b_{i+1} - b_i} \sum_{l=b_i}^{b_{i+1}-1} X_l^{(L)} \right)^2 \tag{5}$$

We further restrict the setting by choosing a maximum group length of $G$ tokens, where we choose $G = 50$ to correspond to one second of tokens, as syllables or words longer than this are fairly rare. With this, we can then split our algorithm into 1) calculating a distance array $D \in \mathbb{R}^{T \times G}$, where $D_{t,g}$ is the cost of the group of length $g$ ending at token $t$ and then 2) solving the minimal interval cover from this distance array with dynamic programming. An efficient implementation using PyTorch (Paszke et al., 2019) on CUDA (NVIDIA et al., 2020) runs in $O(k)$ data-aware sequential steps.

## 4 SYLLABLELM: SPEECH LANGUAGE MODELING ON COARSE UNITS

GSLM (Lakhotia et al., 2021) defines a pipeline for modeling raw audio as three stages: 1) Audio-to-unit Tokenization, 2) Running a language model on these units, and 3) Decoding the tokens back into a waveform. For Audio-to-unit Tokenization, we use the second iteration of SylBoost as described in Section 3. In this section, we lay out our language modeling and resynthesis.

### 4.1 LANGUAGE MODEL

Like AudioLM and TWIST, we use an autoregressive transformer decoder language model to approximate $p(x_t \mid x_{t-1}, \ldots, x_1)$ given an input token sequence $x_1, \ldots, x_T$. We refer to this language model trained on the discrete clusters output by SylBoost as the main SpeechLM. Like TWIST, we prepend a `<BOS>` token and make no other special changes.

### 4.2 TOKEN TO SPEECH DECODING

To convert SylBoost tokens back into audio, we adopt the interleaved decoding strategy from Song et al. (2024) to output the units from TWIST (Hassid et al., 2023), obtaining a waveform by cascading this output into their provided vocoder. This interleaving strategy demonstrates superior performance in high-difficulty settings compared to other text-to-speech models like VALL-E (Wang et al., 2023), and so we use it for all resynthesis experiments. To interleave our units during training, we sort on the start-timestep of every SylBoost unit and TWIST-unit in ascending order. We subtract 0.08s (the length of two TWIST units) from each SylBoost unit start time before sorting to account for noisy SylBoost boundaries. For the rest of the pipeline, we follow Song et al. (2024) without global advance using our syllables as a drop-in replacement for phones and use greedy decoding. We call this model the Interleaved-Vocoder-LM.

Although incorporating this additional resynthesis model slows down generation compared to TWIST, most model scaling happens in the SpeechLM. For example, the TWIST paper still observes scaling improvements in semantic understanding with a SpeechLM of 13B parameters while current SotA speech synthesis models such as Ju et al. (2024) operate with fewer than 1B parameters. Taking the same approach as TWIST, our work focuses purely on improving the semantics of generated audio and is entirely orthogonal to work on audio synthesis quality and voice cloning.

We then generate continuations for a sample by 1) Extracting syllable-units and TWIST units from the sample, 2) Sampling syllable-unit continuations from the SpeechLM, 3) Continuing TWIST units with our interleaved model conditioned on sample TWIST units, sample syllable-units, and continued syllable-units, and 4) Resynthesizing these into speech using the vocoder.

## 5 EXPERIMENTS

### 5.1 TRAINING DATASETS

We train our tokenizer using LibriSpeech (Panayotov et al., 2015), which contains 960 hours of audio books. We noticed that SylBoost converges before all data is used, and so we randomly subsample LibriSpeech to a 100 hour train set and train for five epochs and two iterations for all experiments. We train our SpeechLMs using all of LibriLight (Kahn et al., 2020), which provides roughly 55k hours of speech. As a note on fair comparison, although AudioLM uses exactly this split of LibriLight, TWIST collects an additional 100k hours of data, totaling to 155k hours.

### 5.2 SYLBOOST UNIT CONFIGURATIONS

By varying the number of boundaries input to our cut algorithm at each stage in the SylBoost pipeline, we can arbitrarily control our rate of temporal tokenization. We evaluate three main unit rates at 8.33Hz, 6.25Hz, and 5.00Hz, the latter which matches the empirical rate of SD-HuBERT units on LibriSpeech dev-clean. To account for differences between slow and fast speakers, we dynamically choose $k$ during SylBoost to best match the number of distinct feature groups in the embedding space (seen as blocks in the feature self-similarity matrix $A$, Figure 2). We do this by setting a threshold $\delta$ and taking the fewest boundaries $k$ such that the extraction objective (Equation 5) per timestep is less

Table 1: Unsupervised syllable boundary detection and clustering accuracy on LibriSpeech (Panayotov et al., 2015) test. For F1 scores, the superscript is tolerance threshold in ms. All other metrics use 50ms. Higher is better.

| Approach | Backbone | Training | F1$^{50}$ | F1$^{20}$ | Pr. | Re. | R | CPur | SPur |
|---|---|---|---|---|---|---|---|---|---|
| Feat-Sim(Peng et al., 2023) | HuBERT | no | 47.3 | 24.7 | 46.6 | 48.0 | 54.5 | 28.0 | 30.0 |
| LossPred 5.0Hz (Ours) | HuBERT | no | 59.6 | 31.4 | 54.9 | 66.7 | 56.3 | - | - |
| SD-HuBERT(Cho et al., 2024) | HuBERT | yes | 66.1 | 32.2 | 64.9 | 67.4 | 70.7 | **43.2** | 45.0 |
| SylBoost 5.0Hz (Ours) | HuBERT | yes | 70.9 | 40.1 | 70.8 | 71.4 | 75.1 | 28.9 | 47.8 |
| SylBoost 5.0Hz (Ours) | Data2Vec2 | yes | **73.2** | **44.6** | **72.1** | **74.4** | **76.9** | 33.6 | **54.9** |

Table 2: Boundary detection with different initialization using HuBERT on LS dev-clean

| Model | F1 | Pr. | Re. |
|---|---|---|---|
| Feat-Sim | 46.7 | 48 | 45 |
| -Iter 1 | 51.1 | 50 | 52 |
| -Iter 2 | 50.4 | 51 | 50 |
| LossPred | 60.1 | 53 | 68 |
| -Iter 1 | 67.1 | 67 | 68 |
| -Iter 2 | **70.2** | **70** | **70** |

Table 3: Controllability of unit rate measured on LibriSpeech dev-clean boundary detection. D2V2, 50ms threshold. P:Phone, S:Syllable, W:word

| Hz | F1-P | F1-S | F1-W |
|---|---|---|---|
| 8.33 | **72.0** | 63.5 | 56.8 |
| 6.25 | 65.2 | 71.8 | 66.0 |
| 5.0 | 58.7 | 73.0 | 71.8 |
| 4.3 | 54.3 | **73.2** | **74.0** |

Table 4: Inference speed measured in Real-Time-Factor (RTF), the seconds per processed second. 32 batches with 25s of audio each, 1 GPU, 16 Cores. Hardware in A.4

| Tokenizer Encoder | RTF ↓ |
|---|---|
| SD-HuBERT (Cho et al., 2024) | 2.72E-3 |
| HuBERT+(Peng et al., 2023) | 1.14E-2 |
| HuBERT+Our Extraction (3.3) | **2.05E-3** |
| SpeechLM, 90M, Cached Units | |
| TWIST | 1.28E-4 |
| Ours 6.25Hz 8k | **2.88E-5** |

than $\delta$ for 75% of timesteps. We choose $\delta$ so that average unit rates over LibriSpeech dev-clean match a desired frequency. We also explore modifying the number of clusters generated by K-Means and Agglomerative Clustering, which combined with changing the unit rate results in fine-grained bitrate control. We note that although prior approaches like SD-HuBERT and VG-HuBERT apply a cut algorithm with $k$ cuts, there is no way to control the actual number of distinct feature groups in the self-similarity matrix during training. As a result, we cannot increase the frequency of SD-HuBERT units by changing $k$: Additional cuts result in close-to-identical representations that map to the same quantized clusters.

## 5.3 RESULTS: EVALUATING UNIT QUALITY

We evaluate the quality of our semantic units with two approaches 1) measuring correspondence with syllables and 2) running speech resynthesis followed by ASR. To measure correspondence with syllables, we use the development and test sets of LibriSpeech (Panayotov et al., 2015) and follow the approach from Peng et al. (2023), extracting timesteps for phones using the Montreal Forced Aligner (McAuliffe et al., 2017) and then converting these phones into syllables with a rule-based method (Gorman, 2014). We evaluate the quality of syllable boundary detection with a ground truth boundary marked as hit if a proposed boundary is present within a tolerance threshold. We report F1, Precision, Recall, and R score (Räsänen et al., 2009). We ablate F1 scores with tolerance windows of 20ms and 50ms. Given boundaries, we also evaluate the purity of our clusters at 4096 units. Syllable Purity measures the probability that a syllable is mapped to its most corresponding cluster unit, and Cluster Purity measures the probability that a cluster is mapped to its most corresponding syllable unit.

Even if units do not correspond with syllables, they can still be useful to SpeechLMs if they can resynthesize back into speech that matches the original text. Additionally, training a resynthesis model provides a stronger description of the semantic information contained in units than purity metrics, which may be unreliable because SD-HuBERT does not provide a unit at every timestep while our methods do. To evaluate resynthesized speech, we follow AudioLM and measure Word Error Rate (WER) and Character Error Rate (CER) on the set of 4-10 second segments from LibriSpeech test-clean. For ASR, we follow VALL-E (Wang et al., 2023) and use the public HuBERT-base CTC ASR model provided by Hsu et al. (2021).

Table 1 shows our syllabic correspondence results against the prior-state-of-the-art SD-HuBERT (Cho et al., 2024) and the HuBERT-based feature similarity strategy from Peng et al. (2023). Applying SylBoost to either HuBERT or Data2Vec2 improves correspondence across-the-board except for

Table 5: Unit Resynthesis. WER/CER results on 4-10 second examples on LibriSpeech (Panayotov et al., 2015) test-clean. Hz and Bitrate are measured post deduplication on LibriSpeech dev-clean.

| Model | Changes | Hz | #Units | BPS | WER↓ | CER↓ |
|---|---|---|---|---|---|---|
| SD-HuBERT (Cho et al., 2024) | | 5.0 | 4096 | 60 | 37.3 | 22.7 |
| SylBoost (HuBERT) | +Our Clustering | 5.0 | 4096 | 60 | 18.5 | 10.2 |
| SylBoost (Data2Vec2) | +Use Data2Vec2 | 5.0 | 4096 | 60 | 12.8 | 6.4 |
| SylBoost (Data2Vec2) | +Increase #Units | 5.0 | 16384 | 70 | 9.1 | 4.3 |
| SylBoost (Data2Vec2) | +Tune unit-rate, #Units | 8.33 | **2048** | 91 | 8.0 | 3.7 |
| SylBoost (Data2Vec2) | +Tune unit-rate, #Units | 6.25 | 8192 | 81 | **7.0** | **3.2** |
| TWIST Tokenizer (upper bound) (Hassid et al., 2023) | | 19.5 | 500 | 175 | 6.3 | 2.5 |

in cluster purity. Although LossPred trails SD-HuBERT in performance, it pushes the boundary for syllable recognition using HuBERT without additional training. Improvement across iterations and with different loss initialization can be found in Table 2, which shows that using LossPred as a bootstrapping source instead of HuBERT similarity (Peng et al., 2023) makes SylBoost converge to much higher segmentation accuracy. We discuss initialization more in Appendix A.2. In Table 3, we demonstrate the controllability of SylBoost to extract boundaries at different granularities: higher unit rates successfully correspond more with phones and lower unit rates correspond more with syllables and words. In Table 4 we show the speedup of our efficient unit extraction.

We demonstrate the step-by-step changes used to improve unit cluster re-synthesis quality as compared to SD-HuBERT in Table 5. We observe over a 50% decrease in WER and CER by using SylBoost with the same clustering parameters as SD-HuBERT. We further decrease WER by a third by using Data2Vec2, and from there by modifying the unit sample rate and number of clusters can reach as low as 2048 clusters and a WER of 7%. These results demonstrate by far the lowest bitrate we are aware of for "reasonable-quality" self-supervised-unit resynthesis. Resynthesis for all models we train is done back into TWIST-Tokenizer units, bounding potential quality at a WER of 6.3%.

## 5.4 SPEECH LAUGUAGE MODEL CONFIGURATION

All of the SpeechLMs we implement, as well as our Interleaved-Vocoder-LM, follow the OPT (Zhang et al., 2022) architecture and default to using 12 Transformer layers, an embedding dimension of 768, and learned positional embeddings. This totals to 90M non-embedding parameters, and represents an identical model architecture to TWIST-125M. We also experiment with a larger 24 layer 1024 dimension model totaling to 300M non-embedding parameters, the same as AudioLM and TWIST-350M. For all language model pretraining experiments, we randomly crop files to 25 seconds, use a batch size of 80000 tokens, and train for 200k steps, which amounts to the same compute as in TWIST. To make our approach entirely textless, we do not use TWIST initialization. We tokenize using SylBoost on Data2Vec2 (Baevski et al., 2023), which provided the best results in low-bitrate WER resynthesis (Table 5). We discuss the base encoder further in Appendix A.3. Additional hyperparameters and hardware details are in Appendix A.4.

## 5.5 SPEECH LANGUAGE MODEL BASELINES

We compare against a range of SotA SpeechLMs: AudioLM (Borsos et al., 2023), TWIST (Hassid et al., 2023), GSLM (Lakhotia et al., 2021), and the audio-only version of Moshi (Défossez et al., 2024). As a tokenizer, AudioLM uses w2v-BERT tokens (Chung et al., 2021), TWIST operates on tokens from a HuBERT model that the TWIST authors pretrain for an additional iteration on multilingual data, Lakhotia et al. (2021) uses HuBERT-base and K-Means, and Moshi operates on a residual codec with the first code distilled to match WavLM (Chen et al., 2021). AudioLM and TWIST are the most directly comparable models to SyllableLM with respect to dataset and transformer size and output units at 25Hz followed by consecutive deduplication.

We reimplement a 90M parameter model using the TWIST tokenizer units without textually-pretrained initialization (Cold-Init in the TWIST paper) on our data split for an all-else-held equal comparison on unit type. We additionally reimplement Byte Pair Encoding (BPE) to train a SpeechLM as done in Shen et al. (2024), resulting in the lowest bitrate encoding of speech outside of our model. We

Table 6: Main SyllableLM results. We evaluate on sWUGGY (In-Vocab, All, Out-of-Vocab), sBLIMP from Nguyen et al. (2020), and tStoryCloze from Hassid et al. (2023). Higher is better. *Estimated GPU Hours, see Appx. A.4 for details. Bold is best, underlined is second-best, large models in gray.

| Model | Params | #Units | Hz | BPS | #Data Hours | #Data Toks | GPU-Hours | sWUGGY All | IV | OOV | sBLIMP | tSC |
|---|---|---|---|---|---|---|---|---|---|---|---|---|
| Phone Topline | 90M | 70 | 12.5 | 76 | 55K | 2.5B | 70 | 81.4 | 95.2 | 67.7 | 68.8 | 80.6 |
| Syllable Topline | 90M | 28k | 5.0 | 74 | 55K | 1B | 70 | 79.5 | 93.1 | 65.9 | 69.3 | 76.6 |
| Lakhotia et al. (2021) | 150M | 100 | 50 | 332 | 6K | 1.1B | - | 64.8 | - | - | 54.2 | 66.6 |
| AudioLM (Borsos et al., 2023) | 300M | 1K | 25 | 250 | 55K | 5B | 2.9k* | 71.5 | **83.7** | 59.3 | **64.7** | - |
| TWIST (Hassid et al., 2023) | 300M | 500 | 19.5 | 175 | 155K | 9B | 295 | 70.6 | 80.3 | 61.0 | 56.2 | 69.9 |
| TWIST | 1.3B | 500 | 19.5 | 175 | 155K | 9B | 1.1k* | 71.8 | 81.1 | 62.3 | 57.0 | 70.6 |
| TWIST | 7B | 500 | 19.5 | 175 | 155K | 9B | 5.9k* | 72.7 | 83.6 | 61.8 | 59.0 | 74.1 |
| TWIST | 13B | 500 | 19.5 | 175 | 155K | 9B | 10k* | 73.9 | 84.1 | 63.7 | 59.2 | 76.4 |
| Moshi (Défossez et al., 2024) | 7B | 2K | 12.5×8 | 1.1K | 7M | 2.5T | - | 74.8 | - | - | 59.9 | 80.8 |
| TWIST-CI | 90M | 500 | 19.5 | 175 | 55K | 3.9B | 84 | 69.7 | 79.8 | 59.7 | 55.5 | 69.0 |
| BPE (Shen et al., 2024) | 90M | 4k | 9.8 | 118 | 55K | 2B | 84 | 61.8 | 66.7 | 56.8 | 54.5 | 56.2 |
| SyllableLM | 90M | 2k | 8.3 | 91 | 55K | 1.6B | **70** | **72.2** | 81.7 | **62.6** | 62.4 | 71.4 |
| SyllableLM | 90M | 8k | 6.25 | 81 | 55K | 1.2B | 75 | 72.1 | 82.2 | 61.9 | 62.9 | 70.2 |
| SyllableLM | 90M | 16k | **5.0** | **70** | 55K | **1B** | 82 | 67.6 | 76.9 | 58.3 | 63.2 | 69.0 |
| SyllableLM | 300M | 8k | 6.25 | 81 | 55K | 1.2B | 290 | **72.2** | 82.2 | 62.0 | 63.7 | **75.4** |

Table 7: Holding number of SylBoost units and unit rate constant. ZeroSpeech development set.

| Hz | #Units | sWUGGY | sBLIMP |
|---|---|---|---|
| 8.33 | 4k | **72.9** | 61.8 |
| 6.25 | 4k | 69.3 | **63.3** |
| 5.00 | 4k | 65.7 | 62.8 |
| 8.33 | 2k | 72.1 | **62.0** |
| 8.33 | 4k | **72.9** | 61.8 |
| 8.33 | 8k | **72.9** | 61.2 |

Table 8: Continuation Metrics. We measure PPX@Oracle-VERT and VERT@Oracle-PPX (Described in 5.6)

| Model | PPX@O-V | VERT@O-P |
|---|---|---|
| TWIST 300M | $205 \pm 24$ | $24.0 \pm 1.0$ |
| TWIST 1.3B | $175 \pm 14$ | $22.6 \pm 1.2$ |
| BPE 90M | $148 \pm 12$ | $17.3 \pm 0.9$ |
| 8.33Hz 2k 90M | $159 \pm 8$ | **15.1** $\pm 0.9$ |
| 6.25Hz 8k 90M | $139 \pm 12$ | $20.1 \pm 0.7$ |
| 5.00Hz 16k 90M | 131 $\pm 11$ | 15.2 $\pm 1.0$ |
| 6.25Hz 8k 300M | **116** $\pm 7$ | 15.8 $\pm 0.9$ |

run BPE on deduplicated TWIST units and we grid search to find that the minimum bitrate for BPE occurs at 4k units, which we use for all experiments (Shen et al. (2024) originally operated on 50Hz units, meaning that the 118bps rate obtained here is also new). For textual toplines, we train on corresponding LibriLight text transcripts from Kang et al. (2023) and convert text to phones and syllables using the same methods as in Section 5.3.

## 5.6 RESULTS: SPOKEN LANGUAGE MODELING

The end-to-end GSLM pipeline consists of several components, and so it is essential to have metrics to independently evaluate different stages. To evaluate our SpeechLM stage, we follow Lakhotia et al. (2021) and use the ZeroSpeech (Nguyen et al., 2020) sWUGGY and sBLIMP evaluation. The sWUGGY dataset tasks the model with outputting a higher perplexity on similar but fake spoken words (e.g. brick vs blick). Similarly, the sBLIMP dataset checks syntactic correctness (e.g. the dog sleeps vs the dogs sleeps). We also evaluate the SpeechLM on the tSC set from Hassid et al. (2023), which operates like the ZeroSpeech metrics on a spoken version of the StoryCloze dataset (Mostafazadeh et al., 2016) with the last sentence in negative samples randomly chosen. For all metrics we follow prior work and output the mean perplexity per token.

The results for SpeechLM metrics are depicted in Table 6. We find that training with our syllable units improves performance across-the-board on speech understanding tasks. **With under 90 hours of training, SyllableLM outperforms the 13B parameter TWIST and 7B Moshi on sBLIMP**, which best evaluates semantic understanding (Lakhotia et al., 2021). Measured on all of sWUGGY, we also beat AudioLM with 30x less GPU compute and TWIST model sizes up to 1.3B parameters in lexical understanding. On tSC, we observe that SyllableLM benefits from scaling with our large model approaching the performance of our textual topline. Because our units are much lower frequency, we see a 4.5x speedup at inference time with the same parameter count in Table 4. Due to compute requirements, we are unable to scale further.

We notice a decrease in sWUGGY quality with our 5.0Hz units, which we suspect is in part caused by the short length of the dataset audios making input tokenization excessively short. We further ablate changing the unit rate and unit frequency in Table 7. We also find that BPE, despite having the lowest bitrate outside of our approach, does not approach the quality gains created by our syllable-like units.

To measure the quality of end-to-end continuations, we use the VERT@O-PPX and PPX@O-VERT metrics proposed in Lakhotia et al. (2021), which are shown to be the automatic metrics correlating best with human meaningfulness judgements. VERT@O-PPX measures the diversity of output at the sampling temperature where the perplexity of generated audio transcriptions matches that of the ground truth text, and PPX@O-VERT performs the inverse. Like Lakhotia et al. (2021), we generate 10-second continuations from 1000 randomly sampled 3-second crops from LibriSpeech test-clean, and measure results using their provided environment and parameters. To prevent errors from randomly cutting off audio in the middle of a syllable-like unit, we discard the last encoded unit for all models when generating continuations. We report these in Table 8 with two sigma error bars and find that SyllableLM outperforms TWIST 300M and 1.3B.

# 6  DISCUSSION AND LIMITATIONS

Though speech is a very general medium, there are a number of challenges in adapting our methods to generate low-bitrate units angled towards other audio tasks or other domains such as vision. Our LossPred technique assumes that the semantic units to learn are separable across time, one-dimensional, and contiguous. In audio tasks or settings with multiple speakers, sounds or words can occur simultaneously. Images and video have partially occluded or overlapping objects.

Despite this, LossPred is unique in that it finds boundaries using the values (losses) that the pretrained model is directly trained to optimize. Meanwhile, other approaches such as DINO (Caron et al., 2021) and SD-HuBERT rely on difficult-to-control intermediate features for semantic discovery. SylBoost then demonstrates that we can use LossPred to extract emergent and interpretable features even when they are not clear in the original model. Because of this, we believe that LossPred and SylBoost provide a very promising direction for controllable and scalable semantic feature clustering and extraction in domains like speech, audio, music, and video.

Low-frequency units such as ours also provide a significantly more computationally tractable path toward the scaling that has seen such success in textual language models. On the other hand, SylBoost units may be losing out on useful paralinguistic features like tone whose impact is only salient on non-audiobooks or at scale. We suspect that hybrid models operating on low-bitrate and high-bitrate units could be constructed to balance these considerations and demonstrate significant quality improvements in multimodal speech-text language modeling.

# 7  CONCLUSION

We introduce a new method to tokenize speech for use in SpeechLMs. We do this by proposing a method to elicit syllabic organization in pretrained speech encoder models, bootstrapping a feature-space clustering algorithm from a static analysis of correlations in off-the-shelf SSL encoder model losses across time. We demonstrate the success of our technique both in having strong associations with syllables and as an extremlely low-bitrate codec of speech for textual resynthesis. Using this tokenization strategy, we successfully train SyllableLM, a SpeechLM that out-performs comparable state-of-the-art approaches across a diverse range of metrics with a significant training and inference speedup. We further ablate several design decisions such as quantization strategy, loss initialization, and the effects of controllability for downstream usecases. Compression is a crucial aspect of learning, and we hope that these significant improvements in the unsupervised learning of low-bitrate speech units can serve as a foundation for approaches towards understanding spoken language and general representation learning.

# 8  REPRODUCIBILITY

We focused heavily on reproducibility throughout our work, and we have released all parameters and code used upon deanonymization. We provide the exact equations used to calculate LossPred,

SylBoost, and our Efficient Extraction Pipeline, allowing for reimplementation solely from the main text. The SyllableLM architecture and Interleaved-Vocoder-LM are well-defined and follow publicly sourced prior work. All datasets used for training and evaluation are listed explicitly in the main text, and all evaluation procedures are described as would be needed to calculate them.

## 9 ETHICS

It is important to note that large textual language models can have harmful effects, such as enabling the generation of misinformation in mass. Although generative spoken language models have not yet caught up to their textual counterparts, it is still necessary to be aware of potential misuses that could arise in the future.

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

# A  APPENDIX / SUPPLEMENTAL MATERIAL

## A.1  RANDOMLY SAMPLED EXAMPLE SEGMENTATIONS

We provide randomly sampled example segmentations from the LibriSpeech (Panayotov et al., 2015) dev-clean set. All models are the second iteration of Data2Vec2, which we use for our SyllableLM experiments in Section 5.6. Top: Feature Self-Similarity matrix, darker green is closer. Segmented cuts span vertically in blue from the top, ground truth boundaries span vertically in red at the bottom. Bottom: time-aligned Mel-Spectrogram. We call attention to the interesting behavior of global correspondences appearing when words or syllables are repeated. Best viewed zoomed in.

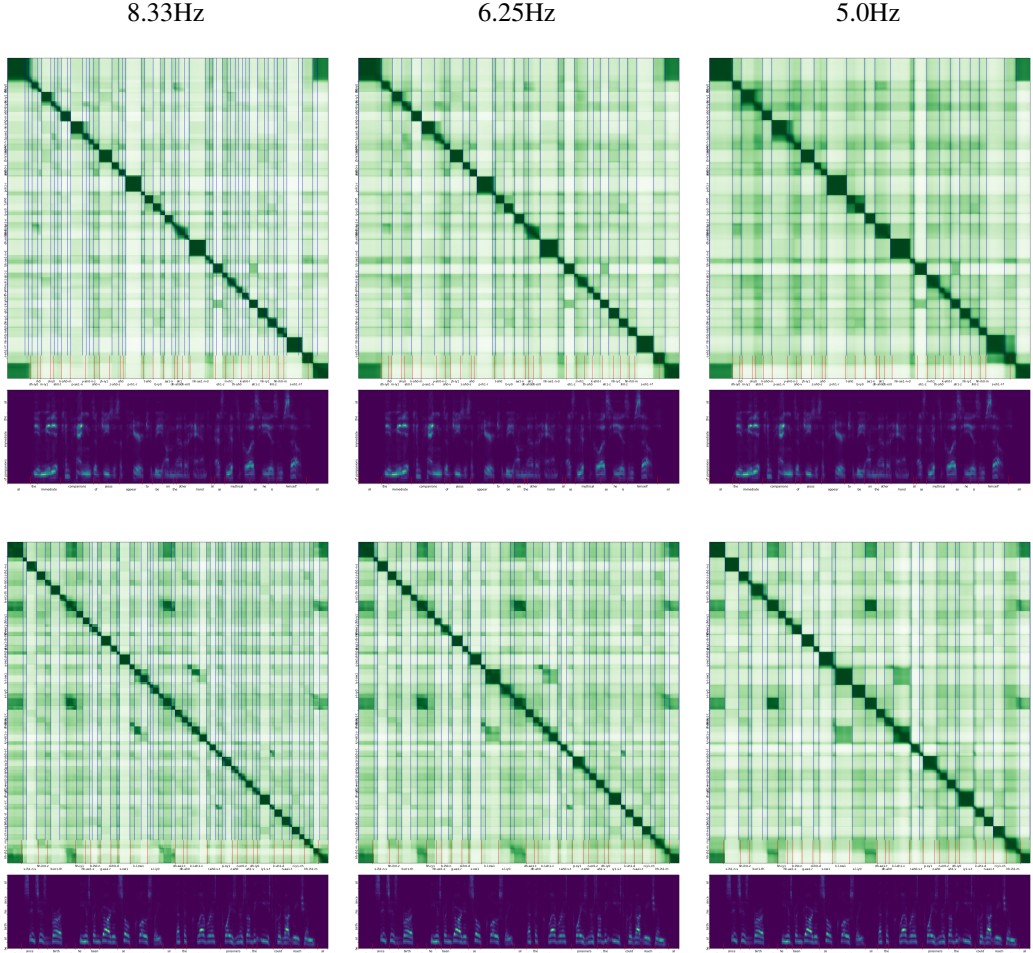

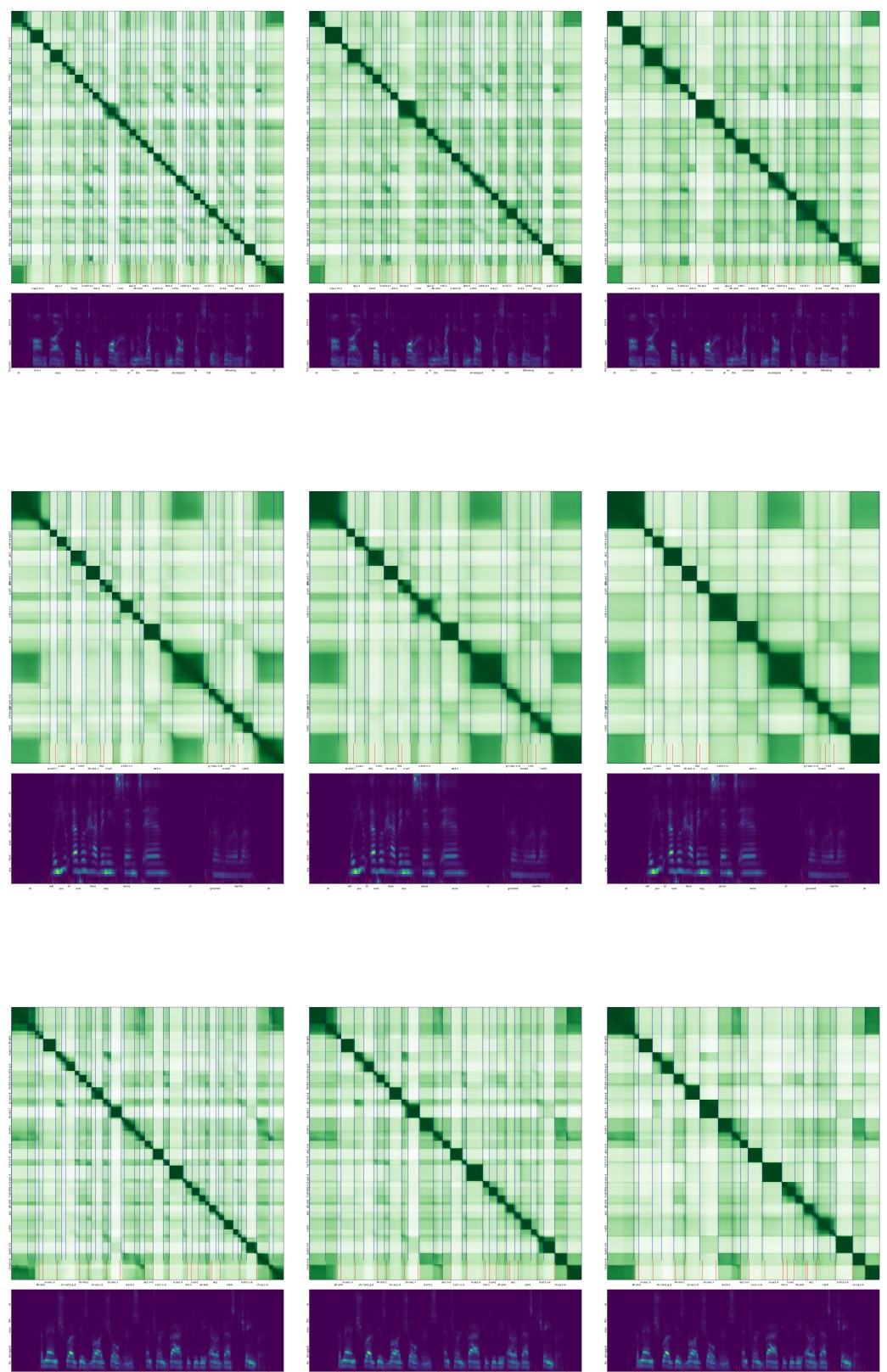

## A.2 DISCUSSION: OTHER BOOTSTRAPPING STRATEGIES

Of course, there already exist several strategies for unsupervised syllable and word segmentation such as in Fuchs & Hoshen (2023), Pasad et al. (2024) and Peng et al. (2023) that could be used to bootstrap our first pseudolabels. We have not conducted an exhaustive search over initializing with these strategies, however it is intriguing that the approach from Peng et al. (2023) achieves significantly lower quality, relatively and absolutely, from SylBoost bootstrapping than LossPred. We suspect that this may be caused by the fact that although the representations of these models correlate with boundaries, there is no modeling in the pretraining loss pushing the representations to linearly separate across semantic differences. Meanwhile, the loss is forced to change across semantic boundaries due to the difficulty of language modeling, albeit noisily.

## A.3 DISCUSSION: BASE ENCODER

Because we want to use a 50Hz base encoder to match SD-HuBERT and have fine-grained boundary control during syllable segmentation, we cannot use the 25hz encoder from TWIST. Unfortunately, this means that the quality of the base encoder may be a confounding factor in our SpeechLM evaluation. We choose Data2Vec2-base (Baevski et al., 2023) as a middleground for training SpeechLMs on syllable-like units because we find its quality enables lower bitrates than HuBERT, but it is older and trains on less-data than the TWIST tokenizer, and it has 6x fewer parameters than w2v-BERT, used by AudioLM. We suspect that applying newer encoders like w2v-BERT 2 from Barrault et al. (2023) could enable even better performance, which we leave to future work. Because the Data2Vec2 loss function doesn't natively work with LossPred, we initialize Data2Vec2 SylBoost from the same HuBERT loss boundaries as discussed in 3.1.

## A.4 HARDWARE AND HYPERPARAMETERS

We implement all experiments using NVIDIA A40 46GB GPUS with a Intel Xeon Gold 6226R CPU @ 2.90GHz. Estimated speeds are made using these results as well as scaling from Zhang et al. (2022).

Hyperparameters for pretraining our models are below. We note that the Batch Size is in terms of tokens, which means that higher unit rates will have fewer seconds of raw audio per batch to keep GPU compute roughly equal per model.

For LossPred, we use a HuBERT-large student and a HuBERT-base teacher as they are the only public checkpoints for a corresponding student and teacher model. We preprocess the audio input to LossPred and Feature Similarity (Peng et al., 2023) using an unsupervised voice activity dection model (Tan et al., 2020).

Table 9: Speech pre-training hyper-parameters.

|  | SyllableLM Base | SyllableLM Large |
|---|---|---|
| Layers | 12 | 24 |
| Embed Dim | 768 | 1024 |
| MLP Dim | 3072 | 4096 |
| GPUs | 2 | 4 |
| Learning rate | $2 \times 10^{-4}$ | $2 \times 10^{-4}$ |
| Adam $\beta_1$ / $\beta_2$ | 0.9 / 0.98 | 0.9 / 0.98 |
| Weight decay | 0.01 | 0.01 |
| Learning rate schedule | Linear Decay | Linear Decay |
| Dropout | 0.1 | 0.1 |
| LayerDrop | 0.0 | 0.0 |
| Warmup updates | 8,000 | 16,000 |
| Batch size (tokens) | 80,000 | 80,000 |
| Updates | 200,000 | 200,000 |
| Position Embeddings | Learned | Learned |

Table 10: Sylboost Prarameters.

|  | HuBERT Base | Data2Vec2 Base |
|---|---|---|
| $L$ (One-Indexed) | 9 | 11 |
| Learning rate | 5e-5 | 5e-5 |
| Epochs | 5 | 5 |
| LibriSpeech Data | 100 hours | 100 hours |
| Batch Size | 32 | 32 |
| Iterations | 2 | 2 |
| K-Means clusters (before Agglom.) | 16384 | 24576 |

## A.5  ADDITIONAL EXPERIMENTS

Table 11: Relative change in the number of tokens under different audio speed modifications.

| New Length / Original Length | 0.5x | 0.6x | 0.7x | 0.8x | 0.9x | 1.11x | 1.25x | 1.43x | 1.66x | 2x |
|---|---|---|---|---|---|---|---|---|---|---|
| Tokenizer | Speedups ↑ | | | | | Slowdowns ↓ | | | | |
| TWIST | 0.60x | 0.70x | 0.79x | 0.86x | 0.93x | 1.07x | 1.13x | 1.22x | 1.36x | 1.52x |
| SD-HuBERT 5.0Hz 4096 Units | **0.70x** | **0.78x** | **0.84x** | **0.90x** | **0.95x** | 1.06x | 1.11x | 1.21x | 1.32x | 1.46x |
| SylBoost 5.0Hz 4096 Units | 0.34x | 0.52x | 0.75x | 0.87x | 0.94x | **1.03x** | **1.09x** | **1.19x** | **1.25x** | **1.33x** |

We evaluate the robustness of different tokenizations to different audio speeds in Table 11. Changing the speed of audio does not change the content of the words spoken, so we would expect that a tokenizer that captures information content at an abstract level, like syllables or words, would output roughly the same number of units when run on a sped-up or slowed-down version of an audio file. To measure this, we evaluate the relative change in the number of tokens under speedups and slowdowns across all audio files in LibriSpeech dev-clean. We evaluate this for SylBoost, SD-HuBERT, and TWIST-Units. For fair comparison, we evaluate SylBoost and SD-HuBERT under the same settings, with an average unit rate of 5.0Hz and 4096 clusters, as sequential deduplication is affected by the number of clusters.

For speedups, we find that SylBoost performs comparably to SD-HuBERT at light speedups; however, the unit rate collapses under more extreme speedups. Anecdotally, we find that model resynthesis for SD-HuBERT and SylBoost under speedups of 0.7x or higher is largely gibberish, suggesting that these numbers should be considered carefully. For slowdowns, we find that SylBoost is significantly more robust to changes in length.

We additionally run AB human opinion tests to evaluate the grammar, meaningfulness, and diversity of generated continuations, following the Human Meaningfulness-MOS from Hassid et al. (2023). We sample 50 random 10-second continuations from TWIST-1.3B and SyllableLM 6.25Hz 300M, following the approach in Section 5.6, and have five raters evaluate each sample. We find that raters prefer SyllableLM responses 42.2% of the time, TWIST responses 26.6% of the time, and rate results as equally meaningful 31.2% of the time, with a 95% confidence interval of ±5.9%.

## A.6  SAMPLE CONTINUATIONS

Below are sample continuations generated with a temperature sampling parameter chosen to best match Oracle VERT diversity scores. We provide continuations of roughly 3 seconds of audio, sampled randomly from LibriSpeech test-clean. This text is given as output by our HuBERT ASR Model from Hsu et al. (2021), with transcription errors present and with no additional modifications. The source text is bolded, and sometimes cuts off mid-word, which can behave differently per sample based on unit rate and quantization artifacts.

| Model | Continuation |
|---|---|
| TWIST 300M | **HE FILLED FOR AND FOUND THE WIZARD'S BLACK CLOTH THE SQUIRE WAS QU**ITE DEVOTED TO CARRYING IT IN THE PARKS I COULD TELL YOU THAT HE RATHER LOVED ALL THE WIZARDS IN THE MATTER OF HONEST FIGUROUS TEMPERED MEN THEN IMMEDIATELY AND BECAUSE HE DID |
| TWIST 1.3B | **HE FILLET FOR AND FOUND THE WIZARD'S BLACK CLOTH THE SQUIRE WAS QU**ICK IN MOVEMENT A WHEEL OF HIS JOURNEY MADE HIM THE IMPRESSION OF HARE'S BLOOD-SHED THERE WAS NO HELP FOR IT HE SURMISED AND HE DID SO |
| SyllableLM 2k 8.33Hz 90M | **HE FILLED FOR AND FOUND THE WIZARD'S BLACK CLOTH THE SQUIRE WAS** GLAD THAT HIS LAD HAD NOT RETURNED TO KING LENO FOR IT IS PROBABLE THAT HE WOULD HAVE BECOME SO UNWELL THAT HE COULD NOT HAVE SPOKEN |
| SyllableLM 8k 6.25Hz 300M | **HE FILT FOR AND FOUND THE WIZARD'S BLACK CLOTH THE SQUIRE WAS** HALF WAY DOWN THE PATH WITH THE WIZ-ARD BUT HE STOPPED SHORT BEFORE THE GATE CLUTCHED HIS FISTS SEIZED HIS GOWN AND |

| Model | Continuation |
|---|---|
| TWIST 300M | **WHEN THEY WERE OUTSIDE UM SIMPLY LATCHED THE DOOR AND STARTED** UP THE STAIRS AND EVEN WENT INTO THE LITTLE HOUSE AROSE IN THE SETTING SUN SET FORTH THE PLACES WHERE PEOPLE LIVED FROM MON HONEY-SUCKLE HANNEY |
| TWIST 1.3B | **WHEN THEY WERE OUTSIDE UM SIMPLY LATCHED THE DOOR AND STARTED** UP THE TURNPIKE OL HAT DON KILL ME THE SLING DE IN YORN ME WEAVING OUT CHARLEYS SENSE EXAMINED WHAT HE MADE EXAMINES |
| SyllableLM 2k 8.33Hz 90M | **WHEN THEY WERE OUTSIDE UM SIMPLY LATCHED THE DOOR AND STARTED** WALKING IN THEY WERE TOO OLD TO CARE MUCH ABOUT GOING HOME THEIR RELATIVES LEFT |
| SyllableLM 8k 6.25Hz 300M | **WHEN THEY WERE OUTSIDE UM SIMPLY LATCHED THE DOOR AND STARTED** SLOWLY DOWN THE CORRIDOR AND MISSUS BAKER WALKED BESIDE THEODORA THEY WERE NEAR THE OUTER DOOR WHERE |

| Model | Continuation |
|---|---|
| TWIST 300M | **DO BE OR NOT TO BE THAT IS THE QUESTION WHETHER TIS NO**BODY SIBL LINE IN OTHER SHIRTS OR CHOCOLATE NOS MICOTTON BUTTER WHAT WE WERE DO WE SEE THESE HITS WE'VE GOT THE GHOST HERE THEY'RE LOOKING |
| TWIST 1.3B | **DO BE OR NOT TO BE THAT IS THE QUESTION WHETHER TIS NO** GOOD EITHER THAN TO GO THROUGH THE JUDG-MENT OF GAUL AND YOUR DOCTRINES THE LORD YOUR GOD AND YOUR GOSPEL IN RESPECT OF THE POWER OF THIS |
| SyllableLM 2k 8.33Hz 90M | **DO BE OR NOT TO BE THAT IS THE QUESTION WHETHER TIS NO** OTHER THAN ESO'S OWN DESTINY YOU SEE IT IS A LA MISTER PRIOR THAT THIS IS THE CASE |
| SyllableLM 8k 6.25Hz 300M | **DO BE OR NOT TO BE THAT IS THE QUESTION WHETHER TIS NO** REGRET OR NO PLEASURE THAT MAY BE RUSHED INTO ACTION AT ONCE WITH THE GREATEST EAGERNESS OF IMPULSE AND ELASTICITY OF HEART |

| Model | Continuation |
|---|---|
| TWIST 300M | **HE IS CALLED AS YOU KNOW THE APOSTLE OF THE INDI**AN KING WHO IS SO GLORIOUS AND ACTING WHY IS THE OTHER PRINCE NOT BELIEVED BY HIM IN EVERY FAITH THAT IS FREE WILL EXCEPT WHEN HE |
| TWIST 1.3B | **HE IS CALLED AS YOU KNOW THE APOSTLE OF THE INDI**ES SAW WHAT HAD PASSED THROUGH HIM LATER IN ANOTHER BOOK AMONG THOSE WHO HAD ENGRAVED IT THIS VOLUME MISTER PICKWICK THOUGHT IT RIGHT NOT TO INSULT YOU |
| SyllableLM 2k 8.33Hz 90M | **HE IS CALLED AS YOU KNOW THE APOSTLE OF THE IN-VID**ISIBLE BEFORE THEY RECEIVED THE GRACE OF GODAD CHRIST THEN HAD IN THE FAITH OF HIS SON |
| SyllableLM 8k 6.25Hz 300M | **HE IS CALLED AS YOU KNOW THE APOSTLE OF THE INDI**ES HE IS THE FORERUNNER OF TEACHING AND FAR BEYOND IT HE IS THE EXACT SCIENTIST WHO MEASURES THE MOVE |

