# OpenReview forum: "SyllableLM: Learning Coarse Semantic Units for Speech Language Models"
_ICLR.cc/2025/Conference — ICLR 2025 Poster_

### Official Review · Reviewer_p1N2 · 2024-11-01

**Soundness:** 3
**Presentation:** 1
**Contribution:** 3
**Rating:** 6
**Confidence:** 3

**Summary:**

This work introduces a controllable self-supervised technique to merge speech representations into coarser syllable-like units while still preserving semantic information. The propsed method produces controllable-rate semantic units at as low as 5Hz and 60bps and
achieves SotA in syllabic segmentation and clustering. Using these coarse tokens, authors successfully train SyllableLM, a Speech Language Model that matches or outperforms current SotA SpeechLMs on a range of spoken language modeling tasks.

**Strengths:**

Overall, I believe this is a valuable work, featuring a clear motivation, good technical contributions, and impressive experimental results. The use of semantic units as low as 5Hz undoubtedly holds significant value for training SpeechLLMs. Additionally, with under 90 hours of training, SyllableLM outperforms both the 13B parameter TWIST and the 7B Moshi on sBLIMP, indicating that this paper contributes meaningfully to both the speech and multimodal research communities.

**Weaknesses:**

The reason I gave a "boadline rej" is that I still have several concerns due to the somewhat unclear presentation of this paper. If my main questions can be adequately addressed during the rebuttal phase, I would be willing to raise my score.

1) If I understand correctly, SylBoost tokens are extracted from the waveform and include not only a sequence of tokens but also the start and end timestamps for each token. However, it’s unclear how SpeechLLM decodes these tokens when performing speech continuation, particularly in terms of predicting timestamps. Specifically, how exactly are timestamps handled during decoding? Is there a step-by-step process the authors could provide to clarify this? Additionally, the frequent mention of TWIST in Section 4.2 is somewhat confusing. My personal suggestion would be to focus more on the proposed method itself in the methods section, and simply add appropriate citations for references to other works where relevant.

2) I noticed that the authors used a cold-init approach for the 90M model. What are the potential advantages and disadvantages of using a larger text pre-trained model compared to the cold-init approach? How might this affect performance on specific downstream tasks?

3) In addition to semantic information, can SylBoost tokens also model acoustic information? Furthermore, regarding SyllableLM, what specific modifications to SyllableLM might be needed to improve its zero-shot TTS voice cloning capabilities?


Question (3) is intended solely for discussion and does not require corresponding experiments to demonstrate these capabilities.

**Questions:**

See Weaknesses.

**Details Of Ethics Concerns:**

N.A.

---

> ### Author Response · Authors · 2024-11-21
> **Response to Reviewer p1N2**
>
> Thank you for your review. In our response, we hope that we have sufficiently addressed all questions or concerns you proposed.
>
> **Confusion about Resynthesis** Although SylBoost naturally outputs timesteps in addition to a sequence of units, the SyllableLM model uses only the units and ignores the timesteps, matching other SpeechLMs (AudioLM, TWIST, GSLM). Unit durations are used solely for our Interleaved-Vocoder-LM, which uses ground truth timestamps during training, and predicts unit timestamps during inference given only the unit sequence. To achieve this, we adapt the pipeline from Song et al. (2024), "ELLA-V: Stable Neural Codec Language Modeling with Alignment-guided Sequence Reordering," as described in Section 4.2.
>
> Song et al. uses boundaries during training but not at inference, which they describe in detail in their work (We agree with the point you made and so we did not describe their methods in detail: “focus more on the proposed method itself in the methods section, and simply add appropriate citations for references to other works where relevant.”). To make things clear, we will elaborate on the Song et al. pipeline below:
>
> For intuition, consider the VALL-E paper (which we assume you are familiar with). In VALL-E, a language model predicts EnCodec units from phone units, which are then converted into speech using the EnCodec vocoder.
> In a simplified version of the Interleaved-Vocoder-LM, we could directly adapt the VALL-E pipeline by:
>
> 1. Substituting phone units for SylBoost units,
> 2. Replacing EnCodec units with TWIST-units, and
> 3. Decoding TWIST-units back into speech using the TWIST-unit vocoder.
>
> In this setting, no unit timesteps would be necessary for decoding back into raw audio.
>
> However, as Song et al. demonstrated, VALL-E faces robustness issues. To address these, Song et al. proposed the ELLA-V model, which introduces an interleaving strategy to incorporate phone locations during training and predict their positions during inference.
>
> How ELLA-V Works (Simplified):
>
> 1. ELLA-V begins with the first phone from the input and generates the EnCodec units corresponding to that phone.
> 2. Once these units are generated, a special <End-of-phone> token is predicted.
> 3. The next phone from the input is then appended to the token sequence, and the model continues to synthesize the sequence iteratively.
>
> In our adaptation, we follow a similar approach but with these substitutions (described in Section 4.2):
> 1. Replace phones with SylBoost units,
> 2. Replace EnCodec units with TWIST-units, and
> 3. Use SylBoost unit durations (extracted from LibriLight) during training.
>
> This adaptation ensures robust resynthesis while maintaining alignment with the original ELLA-V principles.
>
> We frequently mention TWIST (Hassid et al., NeurIPS 2023) because we use TWIST-units as a resynthesis target. However, we recognize that having an all-caps acronym may be distracting. We welcome suggestions for improving readability while retaining clarity.
>
>
>
> **Cold-Init Approach** We use a cold-init approach for all models we train, including all versions of SyllableLM, TWIST-CI, and Acoustic BPE. This approach aligns with the original task of GSLM, where no textual information is available to the model at any stage.
>
> Our reimplementation of TWIST-90M with Cold-Init achieves sWUGGY/sBLIMP scores of 69.7 / 55.5, compared to the original TWIST paper's reported scores of 68.26 / 53.02 with Cold-Init and 70.75 / 53.92 with Textual-Initialization. While these results are fairly similar, our Cold-Init approach outperforms in sBLIMP, whereas TWIST with Textual-Initialization leads in sWUGGY.
>
> Due to limited compute resources, we did not scale to larger models. For insights into how Textual-Initialization impacts additional downstream tasks, we refer readers to Hassid et al.
>
> **Voice Cloning.** In this work, we focus on high-level semantic tasks rather than low-level tasks like voice cloning. This takes the same approach as Hassid et al., and the Semantic stage of AudioLM (Boros et al., 2023). As demonstrated in AudioLM, lower-level capabilities like voice cloning emerge from the acoustic model stages, which are analogous to the Interleaved-Vocoder-LM. Therefore, following from AudioLM, it is highly likely that we could modify the Interleaved-Vocoder-LM to output EnCodec tokens for zero-shot voice cloning. We do not focus on this aspect in our work because 1) Our primary focus is on improving the high-level task of semantic spoken language understanding from raw audio, 2) We match the TWIST Vocoder for a fair comparison when evaluating continuations, and 3) High-quality voice cloning is well-studied in other work and is orthogonal from our contribution.
>
> If you believe that we have adequately addressed your questions about our work, we would greatly appreciate you raising your score to reflect that.

---

> > ### Comment · Reviewer_p1N2 · 2024-11-28
> > **Response to Author**
> >
> > Thanks for the detailed response from author. I would increase my rating to 6.

---

> ### Comment · Area_Chair_q2RW · 2024-11-25
>
> Dear Reviewer p1N2,
>
> Thank you for your valuable contributions to the review process for the paper! The authors have submitted their rebuttal, and I would greatly appreciate it if you could take a look and provide your response.

---

### Official Review · Reviewer_7QjU · 2024-11-03

**Soundness:** 3
**Presentation:** 3
**Contribution:** 3
**Rating:** 6
**Confidence:** 4

**Summary:**

This paper suggests a discrete unit extraction method for speech, where each unit has a variable length, unlike most discrete audio units with a fixed rate (often 25/50 Hz).
To segment variable length units, they use a pre-trained self-supervised audio model trained on masked language modeling.
They first run the SSL model on the raw sequence to obtain the GT discrete units (teacher predictions).
They then mask a fixed-sized segment and use the SSL model again to measure the likelihood of every teacher token. They repeat this over sliding windows throughout the input, resulting in a square asymmetric “similarity” matrix.
As in previous work, they apply a min-cut algorithm to segment the audio into variable-length units.
They then train a neural model for feature matching, which leverages the produced boundaries in its loss function (average feature teacher feature across the segment).
This is done to make the unit extraction more efficient (resolve the computational-heavy sliding windows) and hopefully generalize better.
To finally derive discrete units, they extract features from the boosted neural model and use a more efficient cut algorithm to segment the sequence.
They then calculate the mean feature over each segment, and cluster using K-means and agglomerative clustering to discretize the units.
Using their novel variable-length discrete units, they train speechLMs and see benefits in several metrics (e.g. sWUGGY, sBLIMP, and tSC).

Edit: I increased my score from 5 to 6 (weak-accept) and the contribution from 2 to 3 (good).

**Strengths:**

- The problem of extracting variable-length representations for audio is important, as the amount of information within different audio segments changes.
- The approach reduces the token rate by x2-3 and SpeechLMs trained on these units obtain better results compared to the fixed-rate VQ/K-means discrete unit extraction.
- The idea of segmenting the audio before clustering is interesting, and seems to be quite efficient.

**Weaknesses:**

Weaknesses:
- The approach for variable-length unit extraction is quite complicated, with a wide range of hyperparameters to control (span size, number of segments, different ways to combine two clustering algorithms, etc).
I’m a bit concerned that this method is overfitted to boost the specific metrics evaluated.  I find the core idea to be grouping before clustering - would you agree?
A simpler algorithm can be: (1) compute a similarity matrix with SSL embeddings (maybe K-band matrix as you did to reduce complexity) (2) segment the audio using the similarity matrix (3) run K-means on the average representation per segment. (no training, no sliding windows)  .

- The metrics calculated to evaluate speechLMs mostly include likelihood ranking and perplexity calculations. However, generative models are mostly used to generate content.
A sample page with audio continuations would help qualitatively check the differences in the generative capabilities of the models. ([1] may be useful to quantify those generative capabilities).
I admit that likelihood ranking is the common practice for evaluating speechLMs, but IMO they don’t measure the thing we’re interested in (I’ll use the model mainly to generate, not to measure likelihood), so evaluating generated outputs is also important and should be added.

- One reason to have variable-rate units is to dynamically distinguish between high-info segments and low-info segments.
In your approach, the number of semantic regions is proportional to the length. In that case, 10 seconds of silence and 10 seconds of high-speed speech would result in the same number of tokens.
The units have a variable length, but each sample has a fixed sample rate. With BPE, in contrast, we might get fewer tokens for 10 seconds of silence compared to 10 seconds of fast speech.

- Table 6: What is the reason for the gray shade on several TWIST? In several metrics, the bolded result is not the best (in contrast to what is stated in the caption). I assume you want to state it is unfair to compare 7/13B models with 300M models (and I agree). In that case, it should be clear from the caption.

- The tokenizer is trained on 960 hours of audio, which got me concerned about scalability in the unit extraction process. Can your algorithm scale so we can compute the units based on (e.g.) 55K hours of data? Can you elaborate on the sample complexity of the process (w.r.t. the length of each sample and the number of samples)

- Table 6: The results suggest BPE (an alternative approach to producing a variable-length representation) degrades the results. This contradicts the findings of other several papers reporting benefits from BPE (additional references attached):
[1] Exploring the Benefits of Tokenization of Discrete Acoustic Units (Interspeech 2024)
[2] On the Effectiveness of Acoustic BPE in Decoder-Only TTS (Interspeech 2024)

- Section 5.2: The explanations on the final unit extraction using clustering are vague. How did you combine K-means and Agglomerative clustering?

- I find the task of unsupervised syllable segmentation is more suitable for speech-oriented venues such as ICASSP and Interspeech.

**Questions:**

Comments:
- It would be interesting to develop a joint algorithm for segmenting and clustering together.
- Table 5: When training your tokenizer on smaller datasets with a single language and low input variability, it is easier to compress them and reduce the bitrate, as you have less information to distill into the discrete units.

Writing/Typos:
Lines 14-17 (abstract) can be simpler and more concise

Line 172: Can you clarify: “C models the raw probabilities of the losses”

Line 158: “Two frozen Hubert models” - you used the same single model, right? (otherwise, their token space wouldn’t match..)

Line 280: TWIST citet->citep

Table 2: Detaction->Detection

line 462: Regarding “the E2E GSLM pipeline is deep” — how about rephrasing it to “consists of several independently trained components”?

---

> ### Author Response · Authors · 2024-11-21
> **Response to Reviewer 7QjU**
>
> **Wide Range of Hyperparameters** We agree with the reviewer that methods that have an excessive number of parameters are susceptible to overfitting and call into question their value. **We strongly believe this is not the case for our work.**
>
> We incorporate three main parameters for extraction, and will explain each:
> * Sliding window size $s$ for LossPred
> * Unit rate (or number of cuts k) shared between LossPred and SylBoost
> * Number of unit clusters
>
> For the sliding window, too short a sliding window would not be sufficient to span all words, and too long a window is more expensive to extract. We choose $s$ to correspond to one second because 1) we have to choose some threshold and 2) we were aware of the complexity of LossPred, and so did not want this parameter to seem overfit or especially selected. As a result, we chose one second as a ‘nothing-up-my-sleeve’ number.
>
> We ablate the unit rate and number of unit clusters extensively throughout our work, and we demonstrate that our approach is compatible with a wide range of parameters in Tables 3, 5, 6, and 7, with various tradeoffs.
>
> The proposed simple algorithm suggested by the reviewer is actually equivalent to FeatSim from Peng et al (2023), which we compare against in Tables 1 and 2. We find that our approach significantly outperforms FeatSim, achieving **+25.9 F1** on Syllable boundary detection and **+19.8 F1** for SylBoost initialization. We hope that these dramatically stronger results versus the simpler FeatSim convince the reviewer that our slightly more complex pipeline is justified. Finally, we note that we are careful throughout evaluation about using validation splits for ablations so as to not overfit to the test set.
>
> **Evaluating Continuations** We quantitatively evaluate the quality of generated audio continuations in Table 8 using the proposed evaluation from GSLM, discussed starting on line 484 in Sec. 5.6. These metrics generate 10-second continuations from 3-seconds of audio and measure their diversity and textual-perplexity according to a textual language model. We find that all our 90M and 300M SyllableLM models outperform TWIST 300M and 1.3B in continuation. We provide transcripts of several example continuations in Appendix A.5, and subjectively believe that SyllableLM demonstrates significantly better performance.
>
> **Dynamic unit rates.** We address that we implement a dynamic number of units in Sec. 5.2, starting on line 315: “To account for differences between slow and fast speakers, we dynamically choose k during SylBoost to best match the number of distinct feature groups in the embedding space.”
>
> Here, we perform your suggested experiment: When we run SyllableLM 5.0Hz on 10,000 random examples from LibriSpeech train, we obtain an empirical unit rate of **4.71Hz**. When we run on the same files, but set to silence, we obtain a unit rate of **1.62Hz**.
>
> **Gray Shade** Yes, we gray the table rows to indicate an unfair comparison. We thank the reviewer for agreeing with this. We will explicitly indicate this in our paper.
>
> **Scalability of unit extraction**: We train on the 960 Hours of audio from LibriSpeech simply because that is what the HuBERT-Base and Data2Vec2-Base models use for pretraining. LossPred only ever needs to be run once for 24 GPU Hours to initialize the first iteration of SylBoost. Additional iterations of SylBoost require the same amount of compute as HuBERT and Data2Vec2, plus or minus a constant factor to compute the loss (less than one transformer layer’s-worth). The HuBERT-Large and Data2Vec2-Large models have both been scaled to 55k hours of speech and 300M parameters, so SylBoost should be scalable to this size as well. We compute SylBoost units on 55k hours of speech to construct the training data for SyllableLM, and this takes roughly 2 GPU Days. We use NVIDIA A40s for all GPU experiments, and further hardware for these numbers can be found in Appendix A.4.
>
> In case you are referring to more classical “Big-O” scalability, all of our methods have the O(n^2) complexity as a standard transformer, although LossPred has a high constant factor. SylBoost extraction is O(k) given CUDA parallelization, which we mention in Sec 3.3.

---

> ### Author Response · Authors · 2024-11-21
> **Response to Reviewer 7QjU (Part 2)**
>
> **BPE Comparison:** Yes, you are correct our results do not show the same trends prior work in BPE. However, all prior work into acoustic BPE we are aware of (“Acoustic BPE for Speech Generation with Discrete Tokens” (ICASSP 2024), "Exploring the Benefits of Tokenization of Discrete Acoustic Units" (Interspeech 2024), "On the Effectiveness of Acoustic BPE in Decoder-Only TTS" (Interspeech 2024)) operate by compressing 50Hz units into smaller representations, obtaining a roughly 2.5-3x reduction in unit duration. For example, your first reference, “Exploring the Benefits of Tokenization of Discrete Acoustic Units" maximally compresses 50Hz HuBERT units by 3.2x into 15.625Hz units, a bitrate of 220bps, and 16384 clusters. **Meanwhile, the TWIST units (without bpe) are 175bps**, 19.5Hz, and have 500 clusters. Our resulting BPE units are 9.8Hz and have 118bps, which is about a **2x compression from prior BPE work**. As a result, we believe that our results from BPE are **not contradictory with prior work, but rather demonstrate that BPE does not necessarily perform better at low bitrates.**
>
> **K-means and Agglomerative clustering** We follow VG-HuBERT and SD-HuBERT for K-Means and Agglomerative clustering and perform K-Means to a large number of clusters and use Agglomerative Clustering to reduce this to a final number of clusters. We choose N=24576 for all versions of K-Means and then perform Agglomerative clustering to the final number of units. We choose 24576 = 3*2^13 as opposed to N=16384 from VG-HuBERT for K-Means because we find that a final output size of N=16384 works well for resynthesis (Table 5), and Agglomerative Clustering requires more clusters input than output. This hyperparameter is not critical. On LibriSpeech dev-clean:
> * N=16384 for K-Means + 4096 Agglom (VG-HuBERT/SD-HuBERT setting) SylBoost 5.0Hz results in a 33.7 CPur, 54.9 SPur
> * N=24576 for K-Means + 4096 Agglom (Our setting) SylBoost 5.0Hz results in 34.1 CPur, 54.9 SPur
>
> **Unsupervised Syllable Segmentation** This is a paper on representation learning for speech. The topics for ICLR explicitly include “applications in audio, speech,” (https://iclr.cc/) so we believe that this paper is appropriate for the ICLR venue. Unsupervised syllable detection is an essential component of our SylBoost pipeline, which obtains SoTA in the compression of raw speech into a **learned representation** that maintains the original text: **We reach a WER of 12.8 at 60bps while the English Language is [40bps](https://www.science.org/doi/10.1126/sciadv.aaw2594). The prior SoTA SDHuBERT has a WER of 37.3 at 60BPS**, and no other existing speech codec we are aware of operates below 118bps. Finally, SyllableLM, which is trained on SylBoost units, obtains SoTA in speech understanding from raw audio, as evaluated in Tables 6,7, and 8.
>
> **Segmenting and Clustering Together** We are hopeful that future work will be able to create better methods to segment and cluster speech, and we deeply believe that SylBoost is a strong starting point for this.
>
> **Datasets and Multiple Languages** We are incredibly interested in applying SylBoost and LossPred to other languages and more complicated settings and hope that future work can address this. We choose our current datasets to match prior work (GSLM, AudioLM, SD-HuBERT, HuBERT, Data2Vec2 all use LibriLight / LibriSpeech).
>
> **“C models the raw probabilities of the losses”** The HuBERT loss is $-log(P(Y|X^M))$. The raw probabilities are $P(Y|X^M)$, without the log. This distinction is visible in Equation 2.
>
> **"Line 158: Two frozen Hubert models - you used the same single model, right? (otherwise, their token space wouldn’t match..)"**
> No, they are different models yet still have a compatible token space. Here is the full referenced excerpt: “LossPred operates using two frozen off-the-shelf HuBERT models, a student model and a teacher model, where the student model has been optimized to predict the quantized representations of the teacher model at layer L, and the teacher model comes from the prior iteration of pretraining as described in Hsu et al. (2021).”
>
> Slightly restated, in the HuBERT paper, a student model is optimized to predict teacher model representations, making their token spaces compatible. We use a HuBERT Large Student and HuBERT Base Teacher (Appendix A.4) (This was not a choice we made, the HuBERT author lost the other checkpoints https://github.com/facebookresearch/fairseq/issues/4042).
>
> Thank you for pointing out the typos.
>
> If you believe that we have adequately addressed your concerns in this rebuttal, we would greatly appreciate you reconsidering your score.

---

> > ### Comment · Reviewer_7QjU · 2024-11-24
> >
> > Thanks for your elaborate response.
> > I decided to change my score from weak reject to weak accept.
> > This was due to several points I got wrong in my initial review:
> > 1. My suggested simplification for the method resembles FeatSim. The two approaches are compared, and the further complication comes with performance benefits.
> > 2. Continuations have already been evaluated.
> > 3. Dynamic unit rates are supported. (10 seconds of silence require less tokens than 10 seconds of speech)
> >
> > Additional requests:
> > 1. Please add the details about the K-means + agglomerative clustering to the main paper.
> > 2. Would you mind sharing the BPE variant's continuation metrics (PPX, VERT in Table 8)?
> > 3. Can you elaborate on your approach to support dynamic unit rates? I want to make sure I got it right.

---

> > > ### Author Response · Authors · 2024-11-25
> > >
> > > We sincerely thank you for listening carefully to our response.
> > >
> > > We will update the provided details about K-Means + agglomerative clustering in our final rebuttal revision. We will do our best to include BPE continuation metrics in our paper, however because the continuation metrics require sampling continuations at many different temperature settings (a process further detailed in GSLM), we may not have the time get these in before the rebuttal deadline. We promise to include these in the camera ready submission, but hope to get them in before the discussion period closes.
> > >
> > > How we support dynamic unit rates: At a high level, our dynamic unit rate selection simply seeks to count the number of blocks in the feature similarity matrix (e.g. Figure 2). Because these blocks aren't exact, we instead use a threshold parameter $\delta$ to control our precision in this selection. In Equation 5, there is a MSE loss for every timestep (timestep $j$, corresponding to feature $X_{j}^L$). This loss indicates the amount to which $X_{j}^L$ belongs to its given cluster group. We search over $k$ such that this loss is less than $\delta$ for 75% of timesteps. This search comes essentially "for free" after calculating the distance array $D$.
> > >
> > > We thank you again for your sincere participation in the review process, and believe that your feedback has made our paper stronger as a result.

---

### Official Review · Reviewer_nVDU · 2024-11-04

**Soundness:** 2
**Presentation:** 3
**Contribution:** 3
**Rating:** 6
**Confidence:** 5

**Summary:**

The authors propose a novel method to extract rough syllabic segment boundaries utilizing loss patterns in HuBERT induced by different input mask spans. With those rough segments, a learning method, SylBoost, is proposed to learn a segmented representation by fine-tuning pretrained SSL models. By controlling the granularity of the initial segments, the authors train multiple models with different frequency of speech segments, ranging from phoneme to word level. Syllables are more accurately and efficiently detected than the previous model, SD-HuBERT. By training LM on those syllable-like units from clustering, the authors suggest SyllableLM and demonstrate that spoken LMs at syllabic granularity can outperform or on par with previous SOTA spoken LMs using smaller model and training data.

**Strengths:**

LossPred and Sylboost are well-motivated, novel ways to learn speech representation beyond phonemes, allowing different model training with different frequency units. The method is meticulously explained. Authors depict real speed up in using these units compared to TWIST. Furthermore, results shown achieve near SOTA results on syllable boundary detection and discovery. Table 5 well ablates their design choices and results shown in Table 6 show clear improvements over existing baselines. In particular, the gain in length of tokens and bitrate is outstanding compared to previous speech unit LMs, which is an important factor in LM.

**Weaknesses:**

There are several concerns regarding some claims/analysis, as detailed below.

1. Lack of verification of syllabic correspondence of higher frequency models: The higher frequency models (6.25 HZ or 8.33 Hz) seem sub-syllabic or representing di- or tri-phones based on Figure 2 and Table 3. For example, there is a huge gain in phoneme boundaries correspondence (F1-P) by changing from 5Hz to 6.25Hz. Though, F1-S is still higher than F1-P, but this would be due to the usage of monophones in evaluation. This evokes a question whether the 6.25Hz GSLM – the best one beating SOTA, is a “syllable” LM. The 5Hz version seems the most syllable-like (and the only evaluated frequency in Table 1), but it is the worst among different frequency versions according to Table 5 and 6, showing high WER and no outstanding performance in LM.  Detailed analysis of the linguistic characteristics of the units at different frequencies, perhaps including examples and a discussion of how they relate to syllables, phonemes, or other linguistic units will make this clear.

2. Overclaiming controllability: The controllability of the segment frequency is given during training not at inference. However, the statement in the abstract, ”Our method produces controllable-rate semantic units” is misleading as it strongly implies that the controllability is given at the inference time. It requires rephrasing or a significant improvement in clarity. Besides, the authors claim a lack of controllability of previous studies, but the mincut approach cited by the authors has knobs that can control the frequency of resulting segments. The previous method may have a controllability though the prior works didn’t leverage the functionality. If true, the claimed novelty on controllability should be reassessed, may be provide additional experiments as evidence; and the manuscript must be accordingly updated.

3. Adaptability on different speaking rate at inference: The audiobook datasets have relatively consistent speaking rate as the speech is very carefully delivered. However, in the real world, people may use very different speaking rates based on contexts or based on accents or dialects. Can a model work on speech with a different speaking rate? For example, if we put speech elongated by two times, would the model output the same number of syllables or doubled? The authors explain that the k is determined by optimizing timestep losses but not clear how this loss is calculated at the inference time.

4. Lack of validation of semantic unit discovery: As the authors are citing Choi et al. (2024), I assume semantics here means concepts, e.g., “dog” is similar to “puppy” though they are phonetically distant. However, syllables by definition are not conveying such conceptual meaning, rather structured phonetic information. While the authors are frequently claiming semantic unit discovery, the paper has no supporting evidence that the discovered units or embeddings are semantics. The low error rates in resynthesis cannot verify whether the feature encodes semantic information. The approach suggested by Choi et al. (2024) may be useful to verify the claims.

5. Lack of valid comparison/evaluation of LossPred: The proposed method for initial boundary extraction, LossPred, is novel and makes a lot of sense. Table 2 shows comparisons using Feat-Sim, but why not VG-HuBERT or SD-HuBERT? Those prior works already showed that the Feat-Sim on the original HuBERT is not accurate in terms of syllabic segmentation. As Feat-Sim works best with VG-HuBERT or SD-HuBERT, a fair comparison should involve Feat-Sim on them, not on HuBERT.

6. Absence of audio quality evaluation of generated speech: The paper severely lacks evaluation of the quality of generated audio which is a very crucial aspect of generative model or codec language model of speech. Though it is not claimed to be the main focus, some quality metrics like Mean Opinion Score (MOS) should be reported.  The WER/CER metrics are not sufficient to show the audio quality.

7. Erroneous calculation of RTF: As defined by sources like https://www.openvoice-tech.net/index.php/Real-time-factor and https://link.springer.com/article/10.1007/s11042-020-10073-7, real time factor is defined as the ratio between the time to process an input over the duration of the input. However, it seems like here, the authors chose to report the reciprocal, which is inadequate. The scores should be corrected to comply with the RTF definition. Also, the RTF comparison over TWIST should be calculated end-to-end, not from cached speech units. This is because RTF is aimed to measure efficiency in “real-world” scenarios, so the calculation should start from raw waveform, simulating real-world inference as much as possible. Moreover, this correction is critical as SyllableLM has an additional layer of segmentation in comparison to TWIST, which is not reflected.

**Questions:**

In the intro, what is evidence for “language models on spoken language still struggle to output semantically meaningful speech.”?

In method, “We apply the SylBoost loss (Equation 4) to model features at layer L, which we select based on syllabic correlation as explored in detail in Pasad et al. (2024).” The cited paper is about words not syllables, where I couldn’t locate a particular mention of syllabic correlation. Please let me know in which part of the reference paper demonstrates it.

1. Provide motivations/reasons for some design choices: Why 75% of timesteps in Section 5.2? Why 8.33, 6.25, and 5.00Hz?

2. On the statement "training a resynthesis model provides a stronger description of the semantic information contained in units than purity metrics”, please provide the rationale for this. It is counter intuitive since resynthesis will be better if the input embedding is closer to low-level, like acoustics. For example, VQ-VAE based models like EnCodec show great performance in resynthesis but are claimed to lack higher-level information like semantics.

3. Typo in Table 2: Boundary detaction —> detection.

4. In Table 6, sBLIMP is designed for evaluating syntax (or grammar), which should not be categorized under “semantics”. Also, sWUGGY-all and tSC are the best in Moshi but bold annotations are erroneously given. Please fix them if the numbers are correct.

---

> ### Author Response · Authors · 2024-11-21
> **Response to Reviewer nVDU (Part 1)**
>
> Thank you for your review. We are glad that you appreciate the clarity of our methods, our SoTA boundary detection results, and SyllableLM’s improvements over existing SpeechLM baselines.
>
> In our rebuttal below, we believe we adequately address each of your concerns.
>
> **1. “Lack of verification of syllabic correspondence of higher frequency models”** The goal of our paper is to create a semantic tokenizer to improve spoken language understanding in SpeechLMs. Toward this, our goal is **not** for all evaluated unit granularities to be equal to syllables. Rather, we name our work “SyllableLM” because we are the first work to make SpeechLMs work on units that capture groups of phones (rather than the subphone units from AudioLM and TWIST), and we show that in the configurations that specifically set the unit rate to be approximately equal to the typical syllabic speaking rate, the units learned by our model correlate well with syllables. Our tokenization approach is entirely unsupervised and never uses ground-truth boundaries for training, so we can’t expect exact correspondence between the units we learn and linguistic concepts like phones, syllables, or words. Nevertheless, both our 5.0Hz units and 6.25Hz units obtain better syllabic correspondence than the prior SoTA SD-HuBERT (Tables 1 and 3), justifying the name “SyllableLM.”
>
> We evaluate all unit durations (5.0Hz, 6.25Hz, and 8.33Hz) on phone boundary correspondence, syllable boundary correspondence, and word boundary correspondence in Table 3 to demonstrate linguistic correspondence. We do not report the non-F1 metrics from Table 1 for 6.25Hz and 8.33Hz units because the unit rate directly affects Precision, Recall, Oversegmentation R-Score, Syllable-Purity, and Cluster-Purity. As a result, our comparisons would not be meaningful as there is not a 6.25Hz or 8.33Hz version of any prior work to compare against. For further intuition on the granularity of units we extract, we point you to Appendix A.1., where we included several examples for phone-segmented syllables and word correspondence for 8.33Hz, 6.25Hz, and 5.0Hz units. Anecdotally, we do see in these examples that the higher frequency models tend to still capture multi-phone segments, even if they do not correspond exactly to syllables. Finally, our 5Hz model is not simply “the worst among different frequency versions.” Our 5Hz model obtains the best performance for any 90M parameter model on sBLIMP and Perplexity@Oracle VERT in Tables 6 and 8, and has the lowest bitrate under a WER of 10% in Table 5.
>
> **2. Overclaiming Controllability** We do not believe that “Our method produces controllable-rate semantic units” is overclaiming controllability. We are **not** asserting that “Our **model** produces controllable-rate semantic units.” Our method to produce controllable-rate semantic units is LossPred+SylBoost, and we rigorously demonstrate that it operates at different unit granularities in Tables 3,5,6, and 7.
>
> **2. Previous Studies Have Knobs** We entirely agree with your concern that prior work has tunable knobs, and **we address this point explicitly in our paper starting on line 320**: “We note that although prior approaches like SD-HuBERT and VG-HuBERT apply a cut algorithm with k cuts, there is no way to control the actual number of distinct feature groups in the self-similarity matrix during training. As a result, we cannot increase the frequency of SD-HuBERT units by changing k: Additional cuts result in close-to-identical representations that map to the same quantized clusters.”
>
> To elaborate further, let’s look at the SD-HuBERT quantization approach, depicted in Figure 2 of the SD-HuBERT paper (Choi et al., 2024):
>
> 1. SD-HuBERT applies a norm threshold on the feature map embeddings, and performs min-cut after this. There is no way to control the number distinct feature groups output by norm thresholding because this is a feature of the emergent embedding space from training.
> 2. After norm threhsolding and applying mincut, SD-Hubert quantizes output features and then applies sequential deduplication to merge repeating units into the same cluster. When we apply additional cuts to a cluster, we find that after a certain point all additional cuts map to the same unit cluster and get merged back together.
>
> To make this point concrete, we calculated the unit rate for SD-HuBERT on 10,000 random examples from LibriSpeech Train. **For SD-HuBERT, with k set to approximate 5.0Hz (default), we get a unit rate of 4.75Hz. With k set to approximate 8.33Hz, we get a unit rate of 4.88Hz**. We hope this sufficiently convinces you that SD-HuBERT is not controllable (VG-HuBERT has similar problems, but we don’t focus on it in this reply because it uses paired speech-image data during training).

---

> ### Author Response · Authors · 2024-11-21
> **Response to Reviewer nVDU (Part 2)**
>
> **3. Adaptability on different speaking rate at inference** We ran your suggested experiment on 10,000 randomly sampled LibriSpeech audios in two settings 1) with the default settings from our paper and 2) slowing down each audio 10% by resampling. In the default setting, we obtain 55.1 SylBoost units per audio on average, and 10% slowed down audios result in 59.2 SylBoost units per audio on average. Despite the duration of the audio increasing by 10%, the number of SylBoost units only increases by around 7.4%.
>
> **3. Lack of clarity on explaining how k is determined.** Thank you for pointing this out. Here, we were referring to our extraction pipeline, which finds the mincut boundaries for the SylBoost Loss (Equation 4) using the student model embeddings at inference time (Equation 5). We will update this in our paper.
>
> **4. Meaning of Semantic Unit Discovery.** Following AudioLM, we use the "Semantic Unit/Token" terminology to describe the highest-level input units in our model. For instance, in AudioLM, semantic units refer to w2vBERT units at 25Hz, which are not aligned with semantic “concepts.” We use this terminology to be consistent with prior work, but agree that in many cases “semantic units” is a misnomer (for example see Kwanghee Choi et al., “Self-Supervised Speech Representations are More Phonetic than Semantic,” Interspeech 2024.)
>
> **4. ”The low error rates in resynthesis cannot verify whether the feature encodes semantic information”** We respectfully disagree. As is common with SpeechLMs (AudioLM, GSLM, TWIST), the data source evaluated is audiobooks. Therefore, virtually all semantic information in the input audio is contained in the text transcripts (the original book). **Successfully reconstructing the text transcripts from our units proves that the units contain this semantic, textual information**. The nuance raised by citing Choi et al. (2024) concerns whether units are linearly separable by known concepts. Our focus in this work is not to learn linearly separable “concept” units but to learn “Coarse Semantic Units for Speech Language Models” (Title). For this, we show our units' semantic capability through downstream language model experiments, achieving lower text perplexity (Table 8) and higher t-Story-Cloze scores (Table 6) than prior methods. We demonstrate that prior SoTA low-bitrate units fail to encode sufficient semantic information, as indicated by a 37% WER for SD-HuBERT.
>
>
> **5. “Lack of valid comparison/evaluation of LossPred”** Our goal in this paper for unit extraction is to extract high-quality semantic units at a controllable rate from raw audio. Our overall unit extraction pipeline requires two algorithms, LossPred and SylBoost. LossPred, a zero-shot algorithm, generates noisy semantic boundaries to bootstrap SylBoost, which requires initialization. As discussed in response to point (2), prior syllabic methods are uncontrollable and unsuitable for initializing SylBoost, making LossPred essential.
> We compare LossPred to FeatSim, as both are zero-shot algorithms based on raw HuBERT models without additional training. In contrast, SD-HuBERT involves extensive training and augmentation, so we compare it to our full SylBoost pipeline, initialized by LossPred. Results show LossPred significantly outperforms prior work in both settings. A direct comparison of LossPred and SD-HuBERT on syllabic correspondence is in Table 1.
>
>
> **6. Absence of audio quality evaluation** We respectfully disagree that providing MOS quality scores is essential to this work. Like TWIST (Hassid et al., NeurIPS 2023), our work focuses solely on ingraining semantic understanding in speech language models on raw audio and does not evaluate or contribute to audio synthesis quality. As demonstrated by VALL-E, high-quality audio generation is achievable from language models with existing neural codecs like EnCodec. While our Interleaved Vocoder LM could output EnCodec units, we prioritized fair comparison in speech continuation metrics by using the same vocoder as TWIST, outputting TWIST-units. This avoids confounding variables from resynthesis errors, necessary for continuation metrics. Admittedly, the TWIST vocoder is fairly robotic. However, replacing it with a high-quality model like VALL-E to measure MOS would be costly and provide little insight into our approach, instead just reaffirming that VALL-E is good, which the research community already knows. Instead, our Interleaved-Vocoder-LM demonstrates compatibility with the broader framework of generating finer-grained tokens, which we agree is critical for SpeechLMs.
>
> **7. RTF** Thank you for pointing out that ASR research uses an inverse definition of RTF. We will happily invert our metrics. We note that we define our inverted RTF explicitly in Table 4, so the inverted metrics we report are consistent and reproducible.

---

> ### Author Response · Authors · 2024-11-21
> **Response to Reviewer nVDU (Part 3)**
>
> **7. Cached Units**: We do not measure end-to-end language modeling results due to scaling limitations. As an academic lab, we lack the compute to train a 7B SyllableLM like Meta’s TWIST. However, in TWIST’s scaling, the tokenizer size remains fixed, making unit extraction time negligible in the 7B SpeechLM regime. To reflect this, we output with cached units and compare tokenizer speeds with other syllabic models in Table 4, finding ours to be SoTA.
>
> **Q: SpeechLMs struggle to be semantic.** The GSLM paper (cited in the following sentence) demonstrates that cascaded ASR+LLM systems significantly outperform speech-only approaches. For example, Table 4 in GSLM demonstrates a 3.12% error on the spot-the-word task using an ASR+LLM system while a HuBERT based SpeechLM gets an error of 31.3%. To some extent, “semantically meaningful speech” is subjective. For this, we point the reviewer to the transcripts of continued speech in Appendix A.5, where it is clear that the TWIST models are still in the N-Gram language model era of generating continuations.
>
> **Q: Syllabic Correlation** Thank you for catching an incomplete citation. Peng et al. (“Syllable Discovery and Cross-Lingual Generalization in a Visually Grounded, Self-Supervised Speech Model” (2023)) implements syllable analysis using the methods from “Comparative layer-wise analysis of self-supervised speech models,” Pasad et al., (2023), built on in “What do self-supervised speech models know about words?” Pasad et al., (2024). We will update our final paper to include these citations.
>
>
> **Q.1. Percent of timesteps:** We tested 50%, 75%, and 90%, finding all clustering metrics within 0.3% F1. Due to space constraints, no table is included. **Unit rates:** 5.00Hz aligns with the empirical rate we observed for SD-HuBERT on LibriSpeech Dev-Clean (post voice-activity-detection), as noted on lines 313-314. Since this is the first work on controllable low-rate units for speech language modeling, we chose 6.25Hz and 8.33Hz ad-hoc, evaluating them thoroughly in Tables 3, 5, 6, 7, and 8.
>
> **Q.2.** You are right that resynthesis does not provide semantic information on its own. Instead, our evaluation focuses on **resynthesis at low bitrates**. As a result, low-level codes that capture acoustics are not applicable. Our units bring down prior SoTA resynthesis at a bitrate of 60bps from 37% WER to a 12.8% WER, and reach a WER of 7.0 at **81bps**. Meanwhile, the lowest bitrate setting of EnCodec is **1500bps**. Resynthesis into text is a stronger baseline for purity metrics because it captures the ground truth information of the original signal (in this case the text of spoken audiobooks) instead of measuring an auxiliary metric such as syllable correspondence, which might be noisy. For example, in Figure 2, the two words “at a” are merged into a single unit by SylBoost 5.0Hz. Merging two stopwords is a completely valid approach to unit quantization that may help textual compression, but would harm syllable or word purity metrics.
>
> **Q.3.** Thanks!
>
> **Q.4.** We believe (and reviewer 7QjU agrees) that it is unfair to directly compare our 90-300M models trained on 55k hours of data versus Moshi and the larget TWIST models, which train 7B-13B transformer models on **155k to 7 Million hours** of data. Despite this, it is incredible that **SyllableLM beats TWIST-7B, TWIST-13B and Moshi 7B on sBLIMP**. We find it valuable to draw this comparison while not detracting from our other results.
>
> If you believe that we have adequately addressed your concerns about our paper, we kindly ask that you consider raising your score.

---

> ### Comment · Reviewer_nVDU · 2024-11-25
> **Response to Rebuttal**
>
> I appreciate the authors’ responses and some of my concerns are resolved. However, two important ones remain.
>
> 1) The response to **6.absence of audio quality evaluation** still does not justify lacking speech quality measures.  First, switching to EnCodec outputs may improve the signal quality, but it does not guarantee that generated sounds sound natural. For example, generated speech can be very broken with inappropriate intonation, putting wrong stresses, erroneously elongating/shortening some words, etc. All these error patterns still can be there with high-fidelity, super-clean audio that EnCodec provides. And this is a known problem in VALL-E style TTS, that the model frequently generates broken prosody or even hallucinates the content. Evaluating subjective (or objective) qualities on naturalness is therefore essential for a generative model for speech. **For the record, the original TWIST reported subjective evaluation given this significance (Figure 3 and Appendix A.7 in TWIST paper).** Secondly, evaluating speech quality is even more important in suggested low-bitrate tokens than other previous speech tokens/codecs, since information is inevitably lost at the cost of having such low-entropy in coding. (Again, speech information is not only about text and a lot of information is tangential to WER.) It is a well-known fact that SSL clusters lack pitch and speaker information, and that is potentially why the TWIST vocoder generates robotic sounds. It remains to be seen from your evaluation whether the generation quality is close to something like TWIST. For instance, duration information can be lost during pooling the features.  Does the token-to-speech generate speech with correct durations? The ASR model could be strong enough to undo those potential errors. These are usually measured by subjective quality evaluations. Since the model is largely dependent on TWIST vocoder, it is important to compare the TWIST vocoder synthesis from HuBERT tokens and that from SyllableLM tokens. The tokens here are not necessarily generated tokens from LM, and they can be some samples from LibriSpeech. Unfortunately, the submission does not include a single audio sample that can be heard and at least gives some sense to the reviewers/readers.
>
> 2) About **3. Adaptability on different speaking rate at inference**, the authors conducted experiments with 10% of increased durations. However, 10% is within the natural variance in speech from the same speaker (in a similar context). However, the speaking rate can be dramatically diverse, based on a speakers accent and or the task (read vs spontaneous etc). As I mentioned in the review, this concern was raised because the dataset tested, LibriSpeech, is an audiobook reading dataset that has limited speaking styles and accents. Therefore, I was curious in the review how speech at very different rates will affect the result. (For the record, I suggested 2 times slower speech.) Moreover, the resulting 7.4% increase in the token length is actually surprising since the input speech was slowed down only by 10%, calling into question the robustness of the method to speaking rate. So, I suggest checking speech with more than 30% difference (slower or faster). Even some preliminary qualitative results such as in A.1 would be helpful.
>
> Given these important aspects that remain unclear, I maintain my original scoring for this paper.

---

> ### Author Response · Authors · 2024-11-25
> **Response to nVDU**
>
> **1. "For the record, the original TWIST reported subjective evaluation given this significance (Figure 3 and Appendix A.7 in TWIST paper)."**
>
> **This is incorrect.** The human evaluation results from TWIST **do not evaluate audio quality.** The metric presented in Figure 3 and Appendix A.7 is a speech "Meaningfulness MOS" (MMOS) that measures the **"grammar, meaning, and diversity"** of speech. This evaluates the naturalness of semantic content, not the perceptual naturalness of the generated audio.
>
> Here is the relevant excerpt from TWIST (Final Paragraph, Page 5, https://arxiv.org/pdf/2305.13009): "Human evaluation. To better assess the quality of the generated outputs, we follow the same setup as the Meaningfulness MOS [MMOS; Lakhotia et al., 2021]. In this setup, human raters are presented with several speech continuations (∼10 seconds) following the same speech prompt (∼3 seconds), and are instructed to evaluate how natural (considering grammar, meaning, and diversity) a given sample is, on a scale between 1–5 with increments of 0.5."
>
> We do not perform the MMOS metric because collecting human meaningfulness scores is expensive (TWIST is funded by Meta). Instead, we measure each evaluated quality using objective metrics from Lakhotia et al. ("On Generative Spoken Language Modeling From Raw Audio") (abbreviated as GSLM): Grammar (sBLIMP), Meaning (t-StoryCloze, Text-Perplexity@Oracle-VERT), and Diversity (VERT@Oracle-PPX)
>
> As a result, we maintain and justify our point that prior top-conference work (e.g., Hassid et al., NeurIPS 2023) has focused on improving speech understanding rather than high-quality generation. Our work focuses on learning semantic units from raw audio to induce semantic understanding akin to how text functions in language models. This is a fundamental problem in speech understanding, such as exploring how children learn to listen and speak before they can read or write (GSLM).
>
> The requested features—intonation, stress, pitch, speaker information, and prosody—are handled by the deep, orthogonal work in high-quality speech generation. Challenges in specific architectures like VALL-E (Wang et al.) have largely been addressed by modern architectures such as NaturalSpeech 3 (Ju et al.). Notably, these TTS systems operate with source units (phones or text) containing **no information** about the original acoustics, yet produce high-quality generation. As stated previously, our Interleaved-Vocoder-LM demonstrates compatibility with these frameworks. Even if resynthesis from our units is initially bad, you can easily make it better for example by adding a speaker embedding to condition on, and then retraining the vocoder and not touching the tokenizer/SpeechLM.
>
> We fully admit that an additional, orthogonal, high-frequency codec is necessary for high-quality generation. We instead argue that SyllableLM is desirable regardless due to scaling (line 289): “most model scaling happens in the SpeechLM. For example, the TWIST paper still observes scaling improvements in semantic understanding with a SpeechLM of 13B parameters while current SotA speech synthesis models such as Ju et al. (2024) operate with fewer than 1B parameters.”
>
> Since audio synthesis quality is not the contribution of our work, we did not include audio samples in our original paper. Instead, we showcased contributions like clustering (Appendix A.1) and continuations (Appendix A.5).
>
> As now requested, **we have set up a demo page for resynthesis at https://syllablelmanonymous.github.io/SyllableLMAnonymous/.** We use the official implementation of the single-speaker TWIST vocoder. The TWIST vocoder sounds robotic because it uses the single-speaker HiFi-GAN (Kong et al., 2020) architecture. For reproducibility, we follow https://github.com/facebookresearch/textlesslib/ and use the model configuration at line 143 of https://github.com/facebookresearch/textlesslib/blob/ba33d669d8284b4f7bfe81e7384e83ab799fe384/textless/checkpoint_manager/__init__.py#L26.

---

> > ### Author Response · Authors · 2024-11-25
> >
> > **"Speech information is not only about text and a lot of information is tangential to WER."** We agree entirely and have never argued otherwise. However, the nuance is that most prior SpeechLM work (e.g., GSLM, AudioLM, TWIST) focuses on spoken audiobooks, which lack diversity. **We agree and explicitly acknowledge in Section 6 (Limitations) that future work should address the task of managing semantic understanding across diverse domains.** That said, research must remain focused to be useful to the community, and this is not our area of contribution. Instead, we demonstrate *extremely* strong results in this well-established setting of modeling the text of spoken audiobooks (**We beat TWIST-13B, Moshi-7B on sBLIMP and TWIST-1.3B in Continuation with SyllableLM-90M in 90 Hours of training**).
> >
> > **Adaptability on different speaking rate at inference** We apologize for misinterpreting your original request for a 2x slowdown. When we perform the adaptability experiment with a 2x slowdown (instead of 10%), we find that SylBoost unit sequences are 1.64x longer on average, not 2x longer. While this is not an exact 1x change, we hope it demonstrates the robustness of our units to changes in audio speed. Additionally, in response to reviewer 7QjU, we showed that silence is encoded at 1.62 Hz compared to the 4.71 Hz of normal speech on LibriSpeech Train using SylBoost 5.0 Hz.
> >
> > We sincerely believe we have addressed all major concerns relevant to our paper's contribution. We respectfully request that, if you agree, you consider revising your score.

---

> > > ### Comment · Reviewer_nVDU · 2024-11-27
> > > **Response to Rebuttal (Part 2)**
> > >
> > > I appreciate the authors for sharing their samples, which sound reasonable. I still believe human evaluation is important to the story here given that the stated goal is to introduce a new tokenization scheme for speech (similar to the introduction of GSLM to spoken language modelling), which need not be at the industrial scale of TWIST, but something to validate the tokens as perceptually preserving the lexical content (and ideally prosodic content, but either result is interesting and important to document). This should be affordable and feasible in the academic setting. These are important aspects that round out this work, and also inform subsequent works in the area, to build upon the findings and limitations here.
> > >
> > > Thank you for reporting the experiment on 2X slowed speed (and the compressed tokenization on silence is also noted), but it's generous to characterize 1.64X increase in token length on 2X slowed speech as _robustness of the units to changes in audio speed_. This is a significant increase that calls to question whether these tokens are syllabic only within a certain range of speaking rate, and not outside of that. Could this be caused by the controlled segment frequency at the training stage? Or what else is going on?

---

> ### Author Response · Authors · 2024-11-27
> **Response to nVDU**
>
> **Human Evaluation** We appreciate your feedback and understand the importance of ensuring comprehensive evaluation metrics. We would like to respectfully point out that your requests regarding human evaluation have changed across your replies. In your first two replies, you asked for “evaluation of the quality of generated audio” and to evaluate that the “generated sounds sound natural.” We pointed out that your main justification for this request (the only point you bolded in your previous reply) was incorrect. Specifically, the TWIST paper does not evaluate naturalness but rather meaningfulness. While this correction was not acknowledged, your most recent reply shifts the focus of your requested human evaluation to primarily requesting a meaningfulness score.
>
> Regarding this newly requested human meaningfulness metric, **we would like to emphasize that this is explicitly addressed in our paper on line 484:** “To measure the quality of end-to-end continuations, we use the VERT@O-PPX and PPX@O-VERT metrics proposed in Lakhotia et al. (2021), which are shown to be the automatic metrics correlating best with human meaningfulness judgements.” **In fact, Lakhotia et al. suggests that the automatic metrics we use are better at isolating and evaluating meaning than human judgement**: "The human results are congruent with the automatic scores, although they tend to prefer more units, perhaps showing that they cannot fully dissociate their judgment of meaning from their judgment of intelligibility."
>
> We believe the current draft sufficiently addresses meaningfulness through these metrics. As stated in our prior reply, we also separately evaluate each quality asked for in meaningfulness judgements, “grammar, meaning, and diversity” (and beat prior work in all of them).
>
> * Grammar: sBLIMP
> * Meanining: tSC, PPX@Oracle-VERT
> * Diversity: VERT@Oracle-PPX
>
> As such, we do not think it is fair to condition acceptance on an additional human-rated metric, which we consider redundant given our existing evaluations and provided qualitative results. While we cannot accommodate this new request before the rebuttal deadline, we commit to including the MMOS scores in the camera-ready version of the paper.
>
> **Adaptability**
> We would like to emphasize that adaptability is a secondary feature of our work, not a core contribution (it is not mentioned it in our abstract, introduction, or core contributions). Regardless of adaptability, we demonstrate a 4x unit duration decrease compared to prior work and achieve significantly stronger speech understanding results.
>
> To constructively discuss the robustness of our approach, we additionally evaluate the robustness of relevant prior work. For this, we have switched to evaluating SylBoost with 4k units, to match SD-HuBERT, to control for sequential deduplication (If there are fewer distinct units, duplicates in a sequence are more likely, and removing these duplicates affects unit rate. Sequential deduplication is performed by AudioLM, TWIST, SD-HuBERT, and our work). We also compare against TWIST, which uses 500 units.
>
> Now, at a 2x slowdown, and still across 10000 files, TWIST (500 units) has a 1.53x unit increase, SD-HuBERT has a 1.51x increase, and SylBoost 5.0Hz, 4k units (Table 1/Table 5 evaluation setting) has a 1.47x increase in the number of units. Therefore, our approach is as robust or more robust against a slowdown in audio than prior work.
>
> We acknowledge that this setting may be limited by using fairly monotonous audiobooks, as discussed in our Limitations. Although we do control for segment frequency at the training stage, this factor cannot be the only affect as we have a lesser increase in unit rate than SD-HuBERT, which does not control for segment frequency during training. We hope that this appeases you with regards to prior work that our approach captures changes in speaking rate equally or better than prior work.
>
> We thank you for participating in a detailed discussion, and would appreciate you raising your score if we have appropriately addressed your concerns.

---

> > ### Comment · Reviewer_nVDU · 2024-11-28
> > **Response to Rebuttal (Part 3)**
> >
> > I thank the authors for considering including human evaluation in the final version. To be clear, the main reason I brought up the speech quality is that this is a new tokenization scheme for speech, and it is important to document how “complete” they are, and what information is retained or discarded in them. This evaluation can involve naturalness in many aspects not supported by automatic evaluations, by measuring subtle and subjective qualities we do not have automatic metrics for. I believe this will be an important addition to this paper.
> >
> > Regarding the adaptability, I appreciate the authors for providing a comparison over the baselines. The gains are marginal, suggesting a lack of adaptability to speaking rate, albeit better than baselines. While the authors have asserted that such adaptability is not the primary goal of the study, the paper is titled SyllableLM implying that the representation is robust to non-uniform and variable speaking rates, in addition to metrics reported in Table 1. The fact that it is better than other tokenization schemes is not sufficient to back the implication that these tokens correspond to syllables, beyond a limited speaking rate regime. That these coarser unites are empirically better for modelling speech is noted and appreciated already.
> >
> > To resolve the above concerns, I suggest the authors provide some additional evidence that these units are robust to speaking rate by performing experiments with diverse speaking rate factors ranging from 50% to 150% (even discounting the 200% result, which may be an extreme) applied to the original speaking rate. If the model outputs syllables robustly, the number of segments should not change more than a moderate fraction from the original. I request that the authors include these results and discussion in the manuscript in the Discussion and Limitations section, as this is an important dimension to validate this technique.
> >
> > Overall, most of my initial concerns are resolved. But the aforementioned caveat in robustness to different speaking rates remains significant. I am willing to raise my score if more compelling evidence of this is provided.

---

> ### Author Response · Authors · 2024-11-30
> **Response to nVDU**
>
> We thank you for being happy with the state of our current meaningfulness evaluation, and we will be sure to include Human MMOS scores in the final version of the paper
>
> **Adaptability**
> We have calculated the robustness metric as suggested, using the evaluation strategy from earlier. Below is the final comparison of SylBoost against other models, highlighting unit rate changes across varying audio duration adjustments. This table will be included in the camera-ready version of the paper, as we are past the revision PDF upload deadline:
>
> |  Model    | 0.5x  | 0.6x  | 0.7x  | 0.8x  | 0.9x  | 1.25x | 1.5x  | 1.75x | 2x    |
> |-------------|-------|-------|-------|-------|-------|-------|-------|-------|-------|
> | TWIST       | 0.59x | 0.70x | 0.78x | 0.85x | 0.93x | 1.12x | 1.24x | 1.39x | 1.53x |
> | SD-HuBERT   | **0.71x** | **0.78x** | **0.82x** | 0.86x | 0.94x | 1.13x | 1.31x | 1.42x | 1.51x |
> | SylBoost    | 0.40x | 0.54x | 0.81x | **0.90x** | **0.96x** | **1.10x** | **1.21x** | **1.29x** | **1.47x** |
>
> To read this table, the closer the unit rate change remains to 1x, the more robust the model is to changes in audio duration.
> * For speedups (<1x): Higher values are better.
> * For slowdowns (>1x): Lower values are better.
>
> For all “non-extreme” speed changes (between 0.7x-1.5x), SylBoost is as or more robust than prior work. SylBoost reflects less than half of the change in unit duration than the change in audio duration from 0.8x-1.75x, which we believe indicates that it is “robust” in these settings. Notably, at a 0.8x, 1.5x, and 1.75x duration change, SylBoost has a fairly reasonable improvement over SD-HuBERT and TWIST.
>
> Interestingly, under extreme speedups (0.5x, 0.6x), SylBoost performs poorly (a 50% change is as extreme as 200% because speed is a multiplier). We suspect that this is due to these speech speeds being significantly out of the training distribution, and as a result should not affect the key takeaways of our technique. Speedups or slow-downs of a factor of 2x are extremely unnatural sounding, so this proposed experiment is likely questionable for those rates. Furthermore, speedups heavily affect other factors such as pitch.
>
> Prior work on speed robustness, such as Kaldi (see [Kaldi GitHub](https://github.com/kaldi-asr/kaldi/)), typically addresses smaller variations in speed (e.g., 0.9x–1.1x, as demonstrated in [this script](https://github.com/kaldi-asr/kaldi/blob/701f13107fda71195ab76a7f9f51ed45ce4ec728/egs/wsj/s5/local/online/run_nnet2_perturb_speed.sh#L39)). Within this standard range, SylBoost demonstrates robust performance, aligning with common evaluation frameworks. Anecdotally, we observe that all models tend to produce near-gibberish resynthesis outputs at 0.5x speed, suggesting that metrics like TWIST and SD-HuBERT unit rates at these extremes should be interpreted cautiously.
>
> During an extreme 2x slowdown, SylBoost outperforms prior work, but results are more similar than the gains SylBoost sees at the more moderate changes at 1.5x.
>
> We hope that this addresses your concerns about adaptability.

---

> > ### Comment · Reviewer_nVDU · 2024-12-01
> > **Response to Rebuttal (Part 4/ Final)**
> >
> > I thank the authors for their responses to my queries and commitments to include suggested analysis in the camera-ready.
> >
> > The new speaking rate analysis an important one to show and discuss in the limitations. Please make sure the reported rates are symmetric about __1X__ ( the presented table is binned differently on either side ?).
> >
> > Taking into account the entire story, I increase my rating.

---

### Official Review · Reviewer_HE4J · 2024-11-04

**Soundness:** 3
**Presentation:** 3
**Contribution:** 3
**Rating:** 8
**Confidence:** 4

**Summary:**

This paper proposes a method to extract speech units from existing pre-trained models such as HuBERT or Data2vec2 at syllable boundaries. They show that they are able to significantly lower the number of units used to encode a particular utterance while maintaining and improving on the quality of re-synthesized speech, or continuation of the speech if used with a language model trained on these new speech units. The method is relatively straight forward and, if it works, allows significant savings in the number of tokens needed to accurately model speech data.

**Strengths:**

- the paper describes the method in sufficient detail to reproduce it
- this appears to be a novel way (although based on much prior work) to discover unit boundaries in speech data, which turn out to be syllables.
- the results are compelling in light of much concurrent work on speech-unit language models, especially as they allow to significantly reduce the cost of training such models due to higher compression ratio

**Weaknesses:**

- while using speech units from HuBERT and other models has been widely explored, one downside of such units is that they tend to mostly capture only the semantic content. Presumably, TWIST units may capture some of the prosodic information, but it is not clear how the authors measure the prosody and expressiveness of this method. An alternative for generative tasks is to simply model speech as text (e.g. via an ASR model) and add additional style vectors. Speech understanding tasks may simply use the pre-trained models which this work bootstraps on directly. It would be very interesting to see some quantitative and qualitative comparisons on this topic.
- "Language models require tokenized inputs" - this is not strictly true depending on how one defines "tokens". While discrete tokens are a popular choice, one can certainly build a language model that uses continuous inputs ("tokenized" by splitting it into distinct representations on the time axis or through methods like what is described in this paper) with discrete or continuous targets on the output.

**Questions:**

see weaknesses section for suggestions on what to add to the paper

---

> ### Author Response · Authors · 2024-11-21
> **Response to Reviewer HE4J**
>
> Thank you for your review. We are glad that you appreciate our paper’s clarity, reproducibility, novelty, and the efficiency of our units.
>
> **Expressiveness** Yes, there is significant prior work on discrete resynthesis from semantic units, with a classic example being Polyak et al. ("Speech Resynthesis from Discrete Disentangled Units"). Units can indeed be highly versatile even without prosody or vocal information (e.g., text tokens or phones). Therefore, it is important to have methods that leverage additional inputs, such as spoken vectors, to condition on expressivity. In this paper, our primary contribution lies in learning semantic units (closer to text or syllables) that improve spoken language modeling. Similar to TWIST (Hassid et al., NeurIPS 2023), our work is orthogonal to the deep and important body of research on resynthesis, which we agree is a highly interesting area. We demonstrate that our SylBoost units are compatible with downstream finer-grained codecs by implementing the Interleaved-Vocoder-LM. However, since we use TWIST-units for output and the TWIST-Vocoder for acoustic resynthesis (necessary for fair comparison on continuation metrics), which is not expressive, we follow Hassid et al. and do not measure expressivity in these results.
>
> **Language models require tokenized inputs** You are correct that there are many strategies for continuous language modeling. Algayres et al. (2023) is a particularly relevant work that we cite, as it explores language modeling with continuous semantic tokens. However, the results from Algayres et al. on continuous modeling (68.5/55.3 sWUGGY In-Vocab/sBLIMP) significantly lag behind discretized approaches, such as traditional HuBERT-50Hz tokens in the same training setting (70.36/53.31) and our approach (82.2/62.9). In the first sentence of our abstract, we use the term "tokenization" in a broader sense, similar to how OpenAI refers to continuous "Vision Tokens." Additionally, our unit reduction technique, whether applied to continuous or discrete tokens, still provides a ~4x speedup in the number of tokens input to downstream models.
>
> Please let us know if there are any additional evaluations you would like us to conduct.

---

> > ### Comment · Reviewer_HE4J · 2024-11-27
> > **response**
> >
> > Thanks for the feedback. I have nothing else to add and will keep my rating.

---

### Author Response · Authors · 2024-11-21
**General Response to Reviewers**

We thank all the reviewers for their thoughtful comments and feedback. We have responded to each reviewer's comments individually below.

---

> ### Author Response · Authors · 2024-11-28
> **PDF Revision**
>
> We have submitted our revised PDF with the following changes
>
> We:
> * Inverted our RTF definition
> * Added a description for K-Means and Agglomerative Clustering
> * Explained that we gray out comparisons against excessively large models (>=7B params) in Table 6.
> * Updated citations for prior correspondence analysis to include Peng et al (2023), Pasad et al (2023).
> * Fixed typos

---

### Meta-Review · Area_Chair_q2RW · 2024-12-26

**Metareview:**

This paper proposes a method to discover coarse, variable-length speech units—roughly corresponding to syllables—by leveraging a two-stage approach. The authors first apply a masking strategy on top of a pre-trained model (e.g., HuBERT/Data2vec2) to estimate segment boundaries from a “loss pattern” perspective. After computing segment-level features, the method clusters them into discrete units, achieving a lower token rate (e.g., around 5Hz to 8Hz), which can be used to train a speech-language model (SyllableLM) achieving better or on-par performance of prior discrete-unit models.

**Strengths** (1) The method provides a novel approach to control the compression rate by learning discrete units to the syllable level. (2) Empirical results show clear improvements on multiple benchmarks (3) The reduced length will improve both training and inference.

**Weaknesses** (1) Some reviewers found the method complex and hard to follow, where the writing can be improved; (2) Some reviewers suggested including qualitative measures (e.g., MOS) and semantic validation of the learned units in the paper, which can strengthen the claim of learned units; (3) Several reviewers also flagged an RTF (real-time factor) calculation issue and recommended measuring true end-to-end efficiency from raw audio to discrete tokens.

**Decision** Overall, the reviewers are positive about the novelty and empirical utility, and the discussion period makes an impact that the reviewers reached a uniform agreement of acceptance.

**Additional Comments On Reviewer Discussion:**

The author resolved the questions of the reviewers and included edits in the updated version. 3 out of 4 reviewers increased their scores which led to a consistent acceptance after the discussion.

---

### Decision · Program_Chairs · 2025-01-22

Accept (Poster)